# Combinatorial Group Testing with Selfish Agents

**Georgios Chionas**
Department of Computer Science
University of Liverpool
Liverpool, United Kingdom
g.chionas@liverpool.ac.uk

**Dariusz R. Kowalski**     **Piotr Krysta**[*]
School of Computer and Cyber Sciences
Augusta University
Augusta, Georgia, USA
{dkowalski,pkrysta}@augusta.edu

## Abstract

We study the Combinatorial Group Testing (CGT) problem in a novel game-theoretic framework, with a solution concept of Adversarial Equilibrium (AE). In this new framework, we have $n$ selfish autonomous agents, corresponding to the elements of the universe $[n] = \{0, 1, \ldots, n-1\}$, and a hidden set $K \subseteq [n]$ of active agents of size $|K| = k \ll n$. In each round of the game, each active agent decides if it is present in a query $Q \subseteq [n]$, and all agents receive some limited feedback on $Q \cap K$. The goal of each active agent is to ensure that its id could be revealed from the feedback as early as possible.

We present a comprehensive set of results for this new game, where we design and analyze adaptive algorithmic strategies of agents which are AE's. In particular, if $k$ is known to the agents, then we show adaptive AE strategies with provably near-optimal maximum revealing time of $O(k \log(n/k))$. In the case of unknown $k$, we design adaptive AE strategies with maximum revealing time of order $n^{k-1}$, and we prove a lower bound of $\Omega(n)$ on the maximum revealing time of any such algorithmic strategies. This shows a strong separations between the two models of known and unknown $k$, as well as between the classic CGT, i.e., without selfish agents, and our game theoretic CGT model.

## 1 Introduction

*Combinatorial Group Testing (CGT)* is a classic area of learning theory, see the book Du et al. [2000]. It is about revealing elements from a hidden set of size $k$, typically much smaller than the size $n$ of the universe of all elements, by asking queries and receiving answers (feedback). Queries are typically subsets of the universe, while feedback is a fixed function that provides some limited information about the intersection of the query with the hidden set, e.g., how big is the size of the intersection. CGT was introduced by Dorfman [1943] in the context of revealing infected individuals in large populations using pooled tests, and has been intensively explored recently during the COVID-19 pandemic, c.f., Augenblick et al. [2020], Mallapaty et al. [2020], Sinnott-Armstrong et al. [2020]. CGT has growing applications to Machine Learning, c.f.: simplifying multi-label classifiers Ubaru et al. [2020], approximating the nearest neighbor Engels et al. [2021], and accelerating forward pass of a deep neural network Liang and Zou [2021]. CGT was also used in data mining and stream processing, c.f.: extracting the most frequent elements Cormode et al. [2003], Cormode and Muthukrishnan [2005], Cormode and Hadjieleftheriou [2008], Yu et al. [2004], Kowalski and Pajak [2022b], quantile tracking Cormode et al. [2005], Gilbert et al. [2002b], Greenwald and Khanna [2001], or approximate histogram maintenance and reconstruction Gibbons et al. [2002], Gilbert et al. [2002a]. Other areas include coding and network communication, c.f.: Kautz and Singleton [1964], Clementi et al. [2001], Porat and Rothschild [2011]. The adversarial aspects of CGT were also considered, c.f., Klonowski et al. [2022], Kowalski and Pajak [2022a].

---

[*]Piotr Krysta is also affiliated with Computer Science Department, University of Liverpool, U.K.

37th Conference on Neural Information Processing Systems (NeurIPS 2023).

We study the Combinatorial Group Testing (CGT) problems in a *novel game theoretic framework*, with a solution concept which is an *Adversarial Equilibrium (AE)*. In this enhanced model, elements are autonomous selfish players, also called agents, while those with id in the hidden set are active. Ideally, the active agents should follow a CGT algorithm to have them revealed after some number of queries have been answered.[2] We study efficient deterministic CGT algorithms which are also AE – that is, which additionally incentivise the active agents to follow the CGT algorithm without any deviations. In particular, they are designed in such a way that an agent could worsen its revealing time if it decides to include/exclude itself from the query created by the CGT algorithm (see Section 2.1 for a detailed definition of AE). We also show substantial limitations on scalability of the maximum revealing time when the number $k$ of active agents is unknown.

**Closest related work and their relevance.** The concept of deterministic Adversarial Equilibrium (AE), used in our work, was introduced in the context of non-adaptive contention resolution (CR) games by Chionas et al. [2023]. CR is a communication problem in which each player has a packet to transmit successfully (i.e., alone) in a given time unit on the multiple-access channel to which it is connected, together with other agents. It was introduced in the context of resolving conflicts in Local Area Networks (LAN), co-funded by the recent Turing Award winner Robert Metcalfe, c.f., Metcalfe and Boggs [1976]. Initially it was assumed that agents would respect the given algorithm. Greenberg and Winograd [1985] proved a lower bound $\Omega\left(\frac{k \log(n)}{\log(k)}\right)$ on the running time of any adaptive deterministic CR algorithm with restricted feedback with collision detection. Capetanakis [1979a,b] and Hayes [1978] independently found an adaptive, deterministic binary tree algorithm to solve the CR problem, which runs in $O(k + k \log(n/k))$ time.

CR could be interpreted as a CGT-type problem with some subtle differences. The main conceptual similarity between CR and CGT is the environment – agents decide to transmit in CR (equivalently, be included in the query in the CGT) or not and they receive feedback from the channel (equivalently, broker of the CGT process – the feedback function). The main, though subtle, difference is that in CR it is required that every active agent needs to eventually be in a singleton intersection with some query (i.e., to transmit successfully on the channel, without interference from other active agent), while this is not required in the CGT – in the CGT, the set of active agents could be revealed based on analyzing the feedback, even if some agents are not in singleton intersections with any queries.

Fiat et al. [2007] initiated the work on CR algorithms in which agents seek to minimize their *individual* latency costs. They designed *randomized* algorithms that are Nash equilibria with bounded latency. Later, Christodoulou et al. [2014] studied CR games where the cost of each agent depends on the number of attempted transmissions before success. As mentioned earlier, Chionas et al. [2023] were the first who introduced *deterministic* game theoretic framework to CR. In particular, they showed non-adaptive adversarial equilibria of well-scalable $O(\log n)$ latency (i.e., revealing time), but only for $k = 2, 3$ active agents. In contrast, we study more general *adaptive algorithms* (which correspond to much richer space of strategies), provide a wider framework for the analysis of adversarial equilibrium for general field of CGT with any feedback function, and design well-scalable equilibria for any size $k$ of the hidden set.

Bolouki et al. [2017] studied a Bayesian game where the population is partitioned in two sets, i.e., healthy and infected individuals. The goal of the former set is to have the infected individuals revealed as soon as possible, while the goal of the latter set is not to get revealed for as long as possible. The only strategy of individuals is to comply or not when they are called for a test. Our setting is quite different in the following aspect. We consider a richer space of strategies, since the agents may choose to comply with the query or to deviate by either including themselves in a query without being called for, or not including themselves in a query while being called for.

**Technical contributions.** Apart from our conceptual contribution of designing an adversarial equilibrium model for general (i.e., adaptive) CGT (see Section 2.1), we demonstrate its difference compared to the classical CGT (and related settings, such as CR) by providing several new techniques and performance bounds, c.f., Table 1. The main conceptual difference between the classical and our CGT setting is that while we preserve all features of the classical model, in addition we augment it

---

[2]Depending on the feedback function, revealing of a hidden agent/element could be direct, when the id of this element is returned in the feedback, or indirect, when the analysis of the feedback for the previous queries ensures that the agent must be in the hidden set.

| Class | (n,k)-CGT | | (n,k)-CGT and (n,k)-AE | |
|---|---|---|---|---|
| | **Lower Bound** | **Upper Bound** | **Lower Bound** | **Upper Bound** |
| known $k$ | $\Omega\left(\frac{k\log(n)}{\log(k)}\right)$ $*$ | $O(k\log(n/k))$ † | $\Omega\left(\frac{k\log(n)}{\log(k)}\right)$ $*$ | $O(k\log(n/k))$ Thm. 3.1 |
| unknown $k$ | $\Omega\left(\frac{k\log(n)}{\log(k)}\right)$ $*$ | $O(k\log(n/k))$ † | $\Omega(n)$ Thm. 4.1 | $O(n^{k-1})$ Thm. 4.2 |

Table 1: Summary of our main results. Asymptotic notations $O, \Omega$ denote an upper/lower estimate wrt a constant factor. All logarithms are to base 2, unless stated otherwise. Results marked $*$ and † were published in Greenberg and Winograd [1985] and in Capetanakis [1979a,b], Hayes [1978], resp.

with the game theoretic aspect. This adds new technical challenges in designing efficient algorithms in this new setting. In the main part, we focus on popular feedback function with collisions, and design two algorithms with two very different revealing mechanisms that are also adversarial equilibria. The first algorithm (see Section 3) assumes knowledge of the number of active agents $k$ and reveals agents' ids one by one, using appropriate shortcuts in the binary search process. The second algorithm (see Section 4.2), designed for unknown $k$, reveals all the active players during the first successful query, by gradually decreasing the size of subsequent queries.

The maximum revealing time of the first algorithm is $O(k\log(2n/k))$, for any $2 \leq k \leq n$. This is very close to the best possible maximum revealing time, due to the lower bound $\Omega\left(\frac{k\log(n)}{\log(k)}\right)$ by Greenberg and Winograd [1985] for *any* adaptive deterministic solution in the considered CGT setting. In contrast, Chionas et al. [2023] designed scalable non-adaptive algorithms only for $k = 2, 3$. Moreover, proving AE for general adaptive CGT algorithms, done in this work, is more challenging than proving it for non-adaptive solutions, as the space of all adaptive algorithms is much bigger and complex than the space of pre-defined sequences of queries, i.e., non-adaptive algorithms.

The existence of such algorithms allows to consider the price of stability, c.f., Nisan et al. [2007], Maschler et al. [2020], in CGT games with respect to adversarial equilibria in adaptive strategies. Our equilibria with deterministic adaptive strategies imply an upper bound of $O(\log k)$ on the price of stability (PoS).[3] This upper bound on PoS follows by dividing our bound $O(k\log(2n/k))$ on the maximum revealing time by the lower bound $\Omega\left(\frac{k\log(n)}{\log(k)}\right)$ of Greenberg and Winograd [1985]. To compare, Chionas et al. [2023] designed adversarial equilibria in less complex deterministic *non-adaptive* strategies in the CR game (related to our setting), achieving constant PoS only for $k = 2, 3$ and the PoS of only $\frac{n}{\Theta(k\log(n/k))}$ for any $k > 3$.

All the abovementioned results and comparisons assume known $k$. In case of unknown $k$, we formally show that this setting is more difficult than the one with known $k$ by proving a lower bound $\Omega(n)$ on the maximum revealing time even if an upper bound $k' \geq 3$ on $k$ is known (see Section 4.1). We also designed a universal AE algorithm for any unknown $k$, mentioned earlier, which has polynomial revealing time if the actual number of active agents is constant in $n$ (see Section 4.2).

**Extensions and applications.** We analyzed several extensions and applications of the new framework, showing its generality. We applied it to other feedback functions: ones that instead of returning an id of a singleton intersection return a "beep" (i.e., a single bit indicating that the intersection is a singleton), and others which return "null" in case of non-singleton intersection of the query with the hidden set. Direct applications include adaptive equilibria contention resolution algorithms and new mechanisms for blockchain. Due to the page limit, all details, proofs and extensions and applications are deferred to the Supplementary Materials, in the form of the full version of our paper.

## 2 The Model, the Problem and the Preliminaries

In *Combinatorial Group Testing (CGT)* types of problems, we are given a universe $[n] = \{0, 1, \ldots, n-1\}$ of $n$ elements and a *hidden subset* $K \subseteq [n]$ of size $|K| = k$, where $2 \leq k \leq n$.[4]

---

[3]PoS is the ratio between the cost of the best-case equilibrium and the cost of the socially optimal outcome. In our case, the cost is the maximum revealing time.

[4]Typically CGT assumes that $k \ll n$, that is, the hidden set is much smaller than the size of the universe, $n$.

Later, when introducing game theoretic framework for CGT, we will also be referring to the hidden subset $K$ as *configuration* of the game. Let $\mathcal{F}_n^k$ be the set of all $k$-element subsets of $[n]$, i.e., the family of all possible hidden sets. The goal of an algorithm is to learn the set $K$ using the smallest number of *queries*. A query $Q \subseteq [n]$ returns a value of some function, called $\mathsf{Feedback}(Q \cap K)$. We consider a classical feedback defined in the following way:

- if $|Q \cap K| = 0$, then $\mathsf{Feedback}(Q \cap K) = \emptyset$ ("empty");

- if $|Q \cap K| = 1$, then $\mathsf{Feedback}(Q \cap K) = i$, where $\{i\} = Q \cap K$ (element $i$ "selected");

- if $|Q \cap K| \geq 2$, then $\mathsf{Feedback}(Q \cap K) = \perp$ ("clash" or "collision" or "2+").

This feedback function is one of the simplest considered in CGT, known in the literature as a ternary feedback or $(2, \log n)$-feedback in the generalized CGT terminology, cf. Klonowski et al. [2022]. If the query has a singleton intersection with the hidden set, then naturally the feedback reveals the respective agent's ID. If the intersection is empty or has size at least 2 (clash), then the feedback is empty or clash, respectively. We also study different feedback functions in extensions, included in the Supplementary Materials due to the page limit.

A (deterministic) algorithm for the CGT problem is an adaptive algorithm which, based on the history of the computation, generates subsequent queries $Q_t$, which are asked by the algorithm in the order specified by index $t$, for $t = 1, 2, 3, \ldots$. Each query $Q_t$ may depend on the feedback $\mathsf{Feedback}(Q_1 \cap K), \ldots, \mathsf{Feedback}(Q_{t-1} \cap K)$ from the previous queries $Q_1, Q_2, \ldots, Q_{t-1}$ generated by the algorithm, where $K$ is the hidden set on which the algorithm is executed. We will refer to the index $t$ as *time*, *step* or *round* of the algorithm. Note that the first query $Q_1$ is hardwired since the feedback history before round 1 is fixed (empty).

**Feedback histories.** Suppose that a given algorithm for a given hidden set $K$, generates a sequence of queries $Q_1, Q_2, \ldots, Q_t$ when the algorithm is executed on $K$ by a round $t$. The corresponding unique sequence of feedbacks received in such execution, $\phi[1..t] = \mathsf{Feedback}(Q_1 \cap K), \mathsf{Feedback}(Q_2 \cap K), \ldots, \mathsf{Feedback}(Q_t \cap K)$, is called a *feedback history* (*of the algorithm on set $K$*, if we know the algorithm and $K$) by round $t$.[5] Note that for a given feedback history there is a unique sequence of queries that resulted in such feedback history – this is because we consider deterministic algorithms. We have $\phi[t] \in \{\emptyset, \perp\} \cup [n]$, for a round $t \in \{1, 2, 3, \ldots\}$, and $\phi[1..t] \in (\{\emptyset, \perp\} \cup [n])^t$ ( when $t$ is irrelevant, we simply write $\phi$, where $\phi \in (\{\emptyset, \perp\} \cup [n])^*$). Although the algorithm is not given set $K$ as an input, for each round $t$ it knows the feedback history. We call a given sequence of queries, or feedbacks, a *valid query sequence/feedback history* iff they are query sequence/feedback history obtained in an execution of the algorithm on some hidden set $K \in \mathcal{F}_n^k$.

**Compatibility.** For a given algorithm, we say that a set $K \in \mathcal{F}_n^k$ is *compatible* with a given feedback history, if the execution of the algorithm on the hidden set $K$ generates that feedback history. For a given feedback history $\phi$, let $\mathcal{K}_\phi \subseteq \mathcal{F}_n^k$ be the family of all sets $K$ compatible with $\phi$, and let $\mathcal{K}_\phi|_i = \{K \in \mathcal{K}_\phi : i \in K\}$.

**Definition 1** (Revealing time). *Suppose we are given an algorithm, a hidden set $K \in \mathcal{F}_n^k$, and an element $i \in K$. Let $\phi_t$ be the feedback history of the algorithm run on the hidden set $K$ for $t$ rounds. The smallest $t$ such that all sets $K' \in \mathcal{F}_n^k$ compatible with $\phi_t$ contain element $i$ is called the* revealing time *of element $i$ on the hidden set $K$.*

Note that, in our feedback model above, the revealing time $t$ of agent $i$ could sometimes result from explicit feedback in round $t$, i.e., $\mathsf{Feedback}(Q_t \cap K) = i$, but in other cases could be deducted implicitly by ruling out non-compatible configurations that do not contain $i$ (c.f., Definition 1).

We say that an algorithm *solves the CGT problem*, or is an $(n, k)$-*CGT*, if for any hidden set $K \in \mathcal{F}_n^k$ the revealing time of any element $i \in K$ is finite. The maximum of the revealing time, over all hidden sets $K \in \mathcal{F}_n^k$ and $i \in K$, is called the *maximum revealing time* of the $(n, k)$-CGT algorithm. A performance measure of a secondary interest is the *maximum completion time*, which is the maximum, over all hidden sets $K \in \mathcal{F}_n^k$, of the minimum round when empty queries follow.[6]

---

[5] $\rho[t_1, \ldots, t_a]$ denotes a sub-sequence of $\rho$ consisting of values of sequence $\rho$ in positions $t_1, \ldots, t_a$.

[6] Clearly, an $(n, k)$-CGT algorithm has to reveal all elements in the hidden set before stopping to generate non-empty queries (feedback to which does not bring any information); therefore, the maximum completion time is not earlier than the maximum revealing time.

## 2.1 Game-theoretic framework for CGT

We now define the game theoretic setting of the CGT problem. The set of available selfish players, also called *agents*, coincides with the elements of the universe $[n]$. The hidden set $K \subseteq [n]$ is the set of *active* agents chosen among available agents, who participate in the given instance of the game. As mentioned earlier, we also refer to the hidden set $K$ as *configuration* of the game. Initially an adversary chooses a configuration $K \in \mathcal{F}_n^k$ and informs every agent of their status with respect to the configuration, i.e., each active agent knows about the fact that it is in set $K$, but does not know ids of the other active agents in $K$.

Each agent then executes the same deterministic algorithm, called a *strategy*, determined by the parameter $n$ (and $k$, if given) and the feedback history. Actions of an agent will also be determined by its unique id. We will usually denote the algorithm as a collection of deterministic algorithms (strategies) $(s_0, s_1, \ldots, s_{n-1})$, one for each of the $n$ agents. In each round, the strategy determines if the agent, say $i$, is in the current query in a round $t$ or not, i.e., $s_i[t] = 1$ or $s_i[t] = 0$, respectively. After each agent decides whether it is present in the query or not, the outcome of the feedback function is computed centrally, based on the combined agents' decisions and communicated to all the agents.[7]

Given an algorithm, the goal of each active agent $i \in K$ is to minimize the revealing time of $i$. Therefore, the *payoff* or *utility* of agent $i \in K$ is a strictly decreasing function of $i$'s revealing time. In order to improve its payoff, an agent $i \in K$ may want to *deviate* from the algorithm at some rounds $t$, in which the corresponding query $Q_t$ generated by the algorithm could be modified by $i$ by either:

- *including* itself in the query $Q_t$: the original query, where $i \notin Q_t$, is modified to $Q'_t = Q_t \cup \{i\}$; or

- *excluding* itself from the query $Q_t$: the original query, where $i \in Q_t$, is modified to $Q'_t = Q_t \setminus \{i\}$.

The agents following the algorithm are not informed about the deviation of the other agent, although could deduct some information later on, based on the received feedback – the feedback is given for the modified query, i.e., $\phi[t] = \mathsf{Feedback}(Q'_t \cap K)$ for the game configuration $K$.

**Adversarial equilibria.** The following notion of an adversarial equilibrium has been introduced by Chionas et al. [2023] in the context of related but more restricted non-adaptive contention resolution games. We extend this definition to the CGT game. A collection of deterministic algorithms (strategy profile) $(s_0, s_1, \ldots, s_{n-1})$, one for each of the $n$ agent, solving the CGT problem, is called an $(n, k)$-*adversarial equilibrium*, or $(n, k)$-AE, iff for any agent $i \in [n]$ and any change of its strategy (a.k.a. a deviation) from $s_i$ to some other strategy $s'_i \neq s_i$ (while all other agents $j \neq i$ follow their strategies $s_j$), if there is a configuration $K$ of $k$ agents for which the change strictly decreases the revealing time of agent $i$, then there is a configuration $K'$ of $k$ agent for which this change strictly increases the revealing time of agent $i$. $K$ ($K'$, resp.) is called an *improving* (*worsening*, resp.) configuration for agent $i$ under deviation $s'_i$. We also call strategies $(s_0, s_1, \ldots, s_{n-1})$ a $n$-AE if it is $(n, k)$-AE for any $k = 2, 3, \ldots, n$.

Essentially, the notion of adversarial equilibrium is similar to the notion of individual Pareto optimality. This means that, for a strategy profile that is adversarial equilibrium, for each agent there does not exist an action (deviation) that improves its payoff in one configuration (compared to its respective payoff before deviation) without worsening it in another configuration (again compared to its respective payoff before the deviation). We refer the reader for a further discussion about this definition to Chionas et al. [2023] and to the Supplementary Materials.

**Universal Adversarial Equilibria (with unknown $k$).** An algorithm is *universal* $(n, k')$-AE if the algorithm is $(n, k)$-AE for every $k \leq k'$. An algorithm which is universal $(n, n)$-AE, is also called universal $n$-AE.

---

[7]Note that, since the algorithm is deterministic, all agents could compute the strategies of other agents – however, due to initially unknown configuration (and, thus, ambiguity) and potential deviations (see the next paragraph), such strategies' computation may not often help the agent to predict the actual feedback (which could help to improve the revealing time by deviation, see again the next paragraph for details).

## 2.2 Preliminaries

**Grim-trigger strategies.** Similarly to the other related models and papers Fiat et al. [2007], Chionas et al. [2023], we will also use the following *Grim-Trigger (GT) mechanisms* in agents' strategies. When the algorithm, run by an agent (who is not deviating), discovers certain state that may suggest deviation of other agents, it (repeatedly) includes itself in subsequent queries for a long period. This is done to discourage agents from creating deviations to the CGT algorithm. We will call such behavior *persistent inclusions (of the agent)*. Strategies with the GT mechanism are called *Grim-Trigger strategies*. The notion of Grim-trigger strategies were also used in a similar context in repeated games, see [Axelrod and Hamilton, 1981, Nisan et al., 2007, Sec. 27.2]. It plays the role of a punishment mechanism, which is necessary for the existence of equilibria.

**Search trees and (ambiguous) queries.** For the purpose of designing algorithms which are AE's, we will study binary trees whose leaves are uniquely labeled with elements of the universe $[n]$ and the internal nodes correspond to queries that could be generated by a search algorithm. More specifically, for such given binary tree and its vertex $v$, a query $Q_v$ contains all leaves in the subtree rooted at $v$. Let $\phi$ be a given feedback history. We say that a query $Q_v$ is $\phi$-*ambiguous* iff there are at least two different configurations $K_1, K_2 \in \mathcal{F}_n^k$ compatible with $\phi$ such that $\mathsf{Feedback}(Q_v \cap K_1) \neq \mathsf{Feedback}(Q_v \cap K_2)$. We could omit parameter $\phi$ whenever it is clear from the context. If the query $Q_v$ associated with vertex $v$ is ambiguous, we also call vertex $v$ ambiguous – all this with respect to some family of configurations $\mathcal{K}$.

**Eulerian tour.** For a given binary tree, we will be often referring to a left-to-right Eulerian tour along the tree (or an Eulerian tour, for short). This is a standard procedure that lists vertices of the tree in the left-to-right recursive order, that is, it lists the root, then recursively vertices in the left sub-tree, followed by recursive enlisting of vertices in the right sub-tree.

## 2.3 Binary tree search algorithms

Tree algorithms have been extensively studied to solve the combinatorial group testing and contention resolution problems. These algorithms can be thought of as a short version of an *Eulerian tour* on the (complete) binary tree in which the leaves correspond 1-1 to the $n$ elements and stations in the literature of CGT and contention resolution, respectively, or agents in our game theoretic framework. As a warm-up, we study the algorithm proposed by Capetanakis [1979a] and we shall call it *Binary Search*, or BS for short. This algorithm is based on a simple divide and conquer technique. Firstly, as said, consider that the total number of agents correspond 1-1 to $n$ leaves of a (complete) binary tree. Let $Q_v$ denote the agents that correspond to the leaves of the subtree with root $v$. We shall call $v$ *vertex with token*, or a *token* for short (we will formalize the appropriate notation in Section 3 where we analyze our more sophisticated algorithm). At each time $t$, agents $Q_v$ form the query, and afterwards $v$ changes its location depending on the feedback $\phi$. Initially, $v$ is the root of the tree and $v$ changes its position as follows. At each round $t$ if $Q_v$ results in no collision, i.e., $\phi[t] \in \{\emptyset\} \cup \{i\}$, where $i \in K$, then token $v$ moves to the next-right subtree where collisions are not yet resolved. If $Q_v$ results in a collision, then token $v$ moves to the left subtree in order to resolve the collision. Essentially, the algorithm BS traverses the *left-to-right Eulerian tour* of the binary tree by doing some shortcuts. In order to understand the shortcuts of the token in the algorithm BS compared to the Eulerian tour see Figure 1(a). Note that if $k = n$, the tour of the token is exactly the Eulerian tour of the binary tree.

BS runs in $\Theta(k + k \log(n/k))$ time. This algorithm was proposed in the classical setting of contention resolution without deviations. It is easy to see that BS is $(n, k)$-CGT but it is not a $(n, k)$-AE. We modify BS by adding the grim trigger strategies as follows. If agent $i$ is not revealed as long as token $v$ is the root of a subtree that it belongs to, it will start including itself in the queries persistently in every following round (we may also set up a cap, say, to these persistent inclusions in the next $4n$ rounds, just to make sure that it is longer than the search algorithm's maximum revealing time without deviation). Let us call this algorithm *Grim Trigger Binary Search* or GT_BS for short. Even with this modification BS is not an $(n, k)$-AE.

**Lemma 1.** GT_BS *is not an* $(n, k)$-*AE.*

In the example in Figure 1(a), in the given configuration, agent 7 could deviate by including itself in the query after the second active agent is selected (i.e., the feedback is equal to the id of this agent). It improves its revealing time compared to the one by following BS. By following this simple strategy,

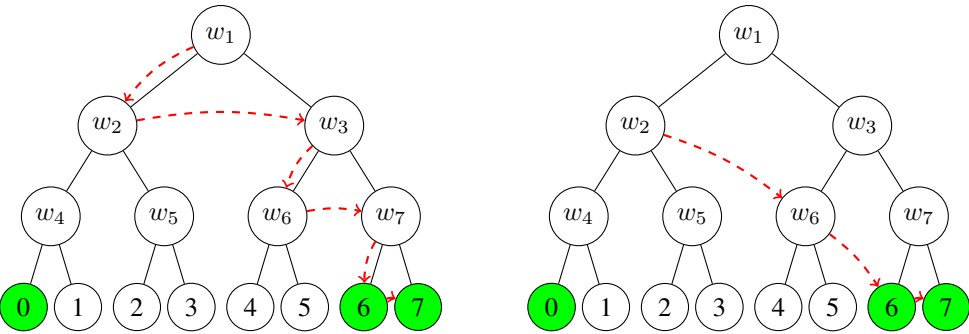

(a) In BS the tour of the token is $(w_1, w_2, w_3, w_6, w_7, 6, 7)$

(b) In BS_Jumps the tour of token, with the appropriate shortcuts, is $(w_2, w_6, 6, 7)$

Figure 1: Here, $n = 8$, $k = 3$, and active agents are $K = \{0, 6, 7\}$. (a): In BS agents are present in the queries with the following sequence according to the token and feedback: $\{0, 1, 2, 3, 4, 5, 6, 7\} \rightarrow \bot, \{0, 1, 2, 3\} \rightarrow 0, \{4, 5, 6, 7\} \rightarrow \bot, \{4, 5\} \rightarrow \emptyset, \{6, 7\} \rightarrow \bot, \{6\} \rightarrow 6, \{7\} \rightarrow 7$. Note that the tour of the token was $((w_1), w_2, w_3, w_6, w_7, 6, 7)$, whereas the Eulerian tour is $(w_1, w_2, w_4, 0, 1, w_5, 2, 3, w_3, w_6, 4, 5, w_7, 6, 7)$. (b): Illustration of BS_Jumps.

the agent will either be better off or will have the same revealing time compared to BS. This leads us to the following important observation that will help us construct our next algorithm (in Section 3): In order for an algorithm to be an $(n, k)$-AE, an active agent who has not been selected should be included in the query immediately after the revealing of the $(k-1)$'st active agent.

## 3    Efficient equilibrium with known size $k$

The algorithm, called *Binary Search with Jumps* (or BS_Jumps, for short), is based on a binary tree, in which leaves correspond 1-1 to agents (this correspondence is part of the input to every agent, thus it is common). More specifically, we enumerate leaves from left to right, using consecutive numbers in $[n] = \{0, 1, \ldots, n-1\}$, and $i$-th leaf corresponds to agent $i$. The tree is complete, if $n$ is a power of 2, or almost complete otherwise, in the sense that all leaves are at the same distance from the root, if possible, or their distances may differ only by 1, respectively.

**Vertex with token and definition of queries.** In each round, an agent maintains a distinguished vertex of the tree, called *vertex with token*, or a *token* for short. The query consists of ids of all leaves that are descendants of the vertex with token. Here a descendant means that the token vertex is located on the path from the leaf to the root, including both ends. From the perspective of an agent, it finds itself included in the query if its corresponding leaf is a descendant of the current vertex with token – we call it an (algorithmic) query inclusion rule.

The initial position of the token, in round 1, is in the leftmost child of the root of the tree. The current (round) position of the vertex with token is computed based only on feedback history and the agent id $i$. The token update rule, described later, is based on the concept of feedback-compatible configurations and ambiguous queries, defined in Section 2 and 2.2, respectively.

**Updating token location.** The location of the token is calculated at the beginning of each round $t$ as follows. Consider an execution of the algorithm by an agent $i$. Let $\alpha_i[t']$ be the location (vertex of the binary tree) of the token at the beginning of step $t'$ of the considered execution, for $1 \leq t' \leq t-1$, stored by agent $i$. We define $\alpha_i[t]$ to be the earliest of the following two vertices on the left-to-right Eulerian tour on the binary tree starting at $\alpha_i[t-1]$:

- the first internal (i.e., non-leaf) vertex $v$ of the tree in that tour such that query $Q_v$ is ambiguous with respect to the set of configurations $\mathcal{K}_\phi|_i$

- the first leaf in that tour that is in every configuration in $\mathcal{K}_\phi|_i$.

Note that if the vertex defined in the first bullet does not exist, it means that all configurations in $\mathcal{K}_\phi|_i$ have the same feedback at each subsequent internal vertex of the Eulerian tour. Then, $\alpha_i[t]$ is

**Algorithm 1:** BS_Jumps$(n, k)$, pseudo-code for active agent $j$

---

$v \leftarrow root$ // Initialization of the token
$K \leftarrow \emptyset$ // Initialization of the set of revealed active agents
**for** $t = 1, 2, \ldots k \log n$ **do**
 **if** $j \in Right(v)$ **then**
  $j \notin Q_v$
 **else if** $j \in Child(v)$ **then**
  $j \in Q_v$
 **else**
  **if** $j \notin K$ **then**
   $j \in Q_v$ // Persistent inclusions if $j$ has not yet revealed
 **if** $\phi[t] = \{i\}$ **then**
  $K \leftarrow K \cup \{i\}$ // Agent $i$ is revealed from feedback $\phi[t]$ in current
   round $t$
 $v \leftarrow update\_token(v, \phi)$
**Output** $K$

---

defined in the second bullet as the first leaf $j$ in the tour such that the corresponding agent $j$ is in every $K \in \mathcal{K}_\phi|_i$.

**Grim-trigger sequence.** If the computed token location $\alpha_i[t]$ is after the leaf corresponding to agent $i$, according to the Eulerian tour, and agent $i$ has not been revealed, it starts including itself persistently in the following queries.

**Termination rule.** The algorithm for agent $i$ finishes its algorithm execution immediately after it is revealed, if it occurs before starting the grim-trigger sequence by this agent. If such revealing does not occur and the agent (eventually) starts its grim-trigger sequence, it continues transmitting until the end of the sequence and then it switches off.

The Algorithm 1 shows how BS_Jumps works. We use three functions:

- $Right(\cdot)$ – which takes as an argument the current token location and returns the set of leaves of a *right* subtree;
- $Child(\cdot)$ – which takes as an argument the token location and returns the set of leaves that are descendants of the token;
- $update\_token(\cdot, \cdot)$ – which takes as arguments the current token location and the feedback history and returns the next token location according to the token update rule (see the two bullets in the "Updating token location" paragraph).

**Efficient implementation of BS_Jumps.** In this work, the main metric of consideration is the revealing time (or query complexity), in which the last agent was revealed. In this paragraph, we briefly discuss other metrics of efficiency of our protocol, such as the computational and communication complexity. The most time-consuming part of algorithm BS_Jumps is the token update. Because of the structure of binary tree, identifying ambiguous internal vertices or leaves that are in every $\phi$-compatible configuration could be done fast by each agent, in each step. In short, the agent simulates locally the Binary-Search-like process and evaluates whether the number of "remaining" leaves is large enough compared to the number of remaining active agents (the latter could be computed from the feedback $\phi$). If it is, it yields ambiguity, as the number of active leaves/agents in the subtree of the currently considered vertex could be $0$ or larger. As this local procedure is built on the top of Binary Search (simulated locally), it is efficient, in the sense that the number of local computational time in any agent is $O(k + k \log(n/k))$, which is $O(1)$ amortized per round. More details of efficient implementation and analysis of correctness and efficiency are given in the Supplementary Material. Regarding the communication complexity of the protocol, it can be derived as a function of the revealing time of the protocol. In each round, each active agent communicates a single bit. The output of the ternary feedback can be encoded by $\log n$ bits. Let $revtime$ be the revealing time of the protocol. Thus, total communication complexity of a protocol is $(k + \log n) \cdot revtime$ bits. The total number of local steps during the protocol could be upper bounded by $\log n \cdot revtime$, because each

step/round of algorithm `BS_Jumps` requires updating of the token, which is a logarithmic operation (see the Supplementary Materials, Section 3). Lastly, the memory needed in each agent is logarithmic, mainly due to the simple recursion in the main algorithm and the token update, and a constant number of variables. This is under assumption that any basic arithmetic operation on variables/values, represented by logarithmic numbers of bits, could be done in a single local step, while these numbers could be stored in a unit of memory.

**Theorem 3.1.** *Algorithm* `BS_Jumps`$(n, k)$ *is an* $(n, k)$-*AE, for any* $2 \leq k \leq n$, *and has revealing time* $O(k \log(2n/k))$.

# 4 Equilibria for unknown size $k$

We first show in Section 4.1 that in the restricted feedback – with collision detection only – universal $n$-AE come at a high cost in performance measures. Intuitively, the high cost in that setting occurs because the adversary has even more power, since it does not only select the configuration but also the size of it and the agents are agnostic to it. We then give a universal $(n, k')$-AE which is an $(n, k)$-CGT with maximum revealing time $O(n^{k-1})$, for any $2 \leq k \leq k' \leq n$. Here, $k$ is the actual number of active agents, unknown to the agents, while $k'$ is the known upper bound. In particular, the maximum revealing time is polynomial for constant-size configurations.

## 4.1 Equilibria characteristics and lower bounds

We start from proving that universal adversarial equilibria require (at least some) dense queries.

**Lemma 2.** *For any* $3 \leq k' \leq n$, *no algorithm with first query of size different from* $n - 1$ *is a universal* $(n, k')$-*AE.*

Intuitively, by applying Lemma 2 to the first $n/2$ queries (assuming that the feedback is collision), we either get impossibility at some point or all these queries have to be of size $n - 1$. This intuition cannot be extended much beyond the first $n/2$ queries produced by an algorithm, because each query of size $n - 1$ somehow restricts the family of configurations of size 2 that we could use in the proof of Lemma 2.

**Theorem 4.1.** *Any algorithm that is an* $(n, k')$-*CGT and universal* $(n, k')$-*AE has the maximum revealing time* $\Omega(n)$, *for any* $k' \geq 3$. *Moreover, the total number of elements in the queries in some executions is* $\Omega(n^2)$.

## 4.2 Equilibrium algorithm for unknown $k$

We now present the algorithm called *Block Builder with Grim Trigger* (or `BB_GR` for short), see Algorithm 2 for a pseudo-code. The structure of algorithm `BB_GR`$(n, k')$, where $k' \leq n$ is the known upper bound on $k$, consists of some $x' \leq k'$ subsequent blocks of queries. In each block of queries $B_x$, for an integer $x$ such that $1 \leq x < x'$, there are $y_x = \binom{n}{n-x} = \binom{n}{x}$ queries, while the last block, $x'$, contains only one query. To simplify the notation for the sake of this algorithm, but with some abuse of the notation used in previous sections, here we denote the $r$th query of block $B_x$ as $Q_r^x$. Until the first selection of an element, we define a query $Q_r^x$ as the $r$-th set from the family $\mathcal{F}_n^{n-x}$ (where the order of these sets could be fixed arbitrarily by the algorithm). When the first successful selection happens, say for some query $Q_r^{x^*}$ in block $x^*$ such that $x^* = x' - 2$ (or $x^* = k' - 1$ in some cases when $x' = k'$, see the second part of the pseudo-code when the value of $x$ has been set to $k' - 1$), the hidden set $K$ is revealed as

$$K = \left\{ \mathsf{Feedback}(Q_r^{x^*} \cap K) \right\} \cup \left( [n] \setminus Q_r^{x^*} \right) .$$

After the first selection, the selected agent with $id = \mathsf{Feedback}(Q_r^{x^*} \cap K)$ includes itself persistently in the following queries up to query $Q_1^{x'}$, which is the first query of the last block $B_{x'}$ executed by the algorithm in the current game. Then, the algorithm stops following the block structure and terminates.

In the analysis of `BB_GR`$(n, k')$ below, we show that the block number of the revealing time, $x^*$, satisfies $x^* = k - 1$, where $k$ is the actual size of a configuration. Consequently, the number of blocks in the game execution is $x' = k + 1$ or $x' = k$, where the latter occurs when $k = k'$ (see the second part of the algorithm's pseudo-code, with these two cases).

**Algorithm 2:** BB_GR$(n, k')$, pseudo-code for active agent $j$

---

**for** $x = 1, \ldots, k' - 1$ **do**

  // We execute blocks $B_x$, until first selection

  **for** $r = 1, 2, \ldots, \binom{n}{n-x}$ **do**

    $Q_r^x \leftarrow r$-th query in $\mathcal{F}_n^{n-x}$ (arbitrarily ordered)

    **if** Feedback$(Q_r^x \cap K) = i$ *for some* $i \in K$ **then**

      $t^* \leftarrow r$

      $K \leftarrow \{i\} \cup ([n] \setminus Q_r^x)$   // Hidden set revealed

      **break**

**if** Feedback$(Q_r^x \cap K) = j$ **then**

  **for** $r = t^* + 1, \ldots, \binom{n}{n-x}$ **do**

    // We continue in block $B_x$ but with grim trigger

    $Q_r^x \leftarrow \{j\}$   // Persistent inclusions

  **if** $x < k' - 1$ **then**

    **for** $r = 1, 2, \ldots, \binom{n}{n-(x+1)}$ **do**

      // We run one more block $B_{x+1}$ with grim trigger when $x < k' - 1$

      $Q_r^{x+1} \leftarrow \{j\}$   // Persistent inclusions

    $Q_1^{x+2} \leftarrow \{j\}$   // Last persistent inclusion when $x < k' - 1$

  **else**

    $Q_1^{x+1} \leftarrow \{j\}$   // Last persistent inclusion when $x = k' - 1$

**Output** $K$

---

**Lemma 3.** BB_GR *is an* $(n, k)$-*CGT, for any* $2 \le k \le k'$*, with maximum revealing time* $O(n^{k-1})$.

**Lemma 4.** BB_GR *is a* universal $(n, k')$-*AE*.

**Theorem 4.2.** BB_GR *is universal* $(n, k')$-*AE with maximum revealing time* $O(n^{k-1})$ *for any number* $k$ *of active agents, such that* $k \le k' \le n$.

## 5 Discussion and further directions

We initiated a study of combinatorial group testing algorithms that are also adversarial equilibria – no agent/element has an incentive to deviate under the presence of an adversary. Our solution in case of known size $k$ of the hidden set is almost optimal, in the sense that the price of stability is $O(\log k)$, i.e., the ratio between our result and the absolute lower bound on revealing time $\Omega(k\frac{\log n}{\log k})$ by Greenberg and Winograd [1985]. Closing this gap, and more importantly, devising an efficient equilibrium for less complex feedback function is an interesting direction. If negative results could be proved for deterministic solutions, then the question about efficient mixed equilibria (i.e., using randomized algorithms as strategies) would be interesting.

Another interesting question would be to try to close the gap between the bounds on maximum revealing time where the number $k$ of active agents is known, that is $O(k \log n)$, and the bounds where $k$ is unknown, which is $O(n^{k-1})$.

Other potential areas of application include: considering different feedbacks (e.g., full quantitative) and other related problems, e.g., discovering hidden multi-sets or transmissions of packets on a multiple-access channel without collisions. One may also consider weaker "risk averseness" in the definition of AE, and how it influences efficiency of CGT.

Finally, it would be interesting to study connections of our games to repeated games, see, e.g., Osborne and Rubinstein [1994], where each agent has to transmit more than one packet.

## Acknowledgments and Disclosure of Funding

The work of Dariusz R. Kowalski was partially supported by the NSF grant 2131538. The work of Piotr Krysta has been partially supported by the Network Sciences and Technologies (NeST) initiative of the University of Liverpool.

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
