*Proof.* Consider $n$ agents, the complete binary tree defined above, and $k$ active agents. Assume that the agent with id $n - 1$ deviates from BSby following the strategy $s'_{n-1}$: after selecting $(k - 1)$-st element, it adds itself to the next query. By playing strategy $s'_{n-1}$, there is no configuration in which the revealing time of the agent is worse compared to the respective one of BS, as after the other $k - 1$ agents have been selected, they stopped being included in subsequent queries and thus there is no active agent to clash with agent $n - 1$ in the round it deviates. However, it is easy to construct a configuration wherein strategy $s'_n$ incurs shorter revealing time, implying an improving configuration, e.g., when agent with id $n - 2$ is not active (i.e., not in this configuration). $\square$

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

**Algorithm 2:** $update\_token(v, k)$

---

$S \leftarrow S \cup \{v\}$
**if** $v = leaf$ **or** $Leaves(v) = 1$ **then**
    ⌊ **return** $v$
**if** $k \geq EulerLeaves(v) - Leaves(v) + 2$ **then**
    ⌊ **return** $update\_token(left\_child(v), k)$
**if** $sibling(v) \in S$ **and** $k \geq 2$ **then**
    ⌊ **return** $update\_token(left\_child(v), k)$
**return** $v$

---

Algorithm 2 uses the following functions:

- $left\_child(\cdot)$: which takes as argument the current token location and returns the root node of its left subtree
- $sibling(\cdot)$: which takes as argument the current token location and returns the node with the same parent
- $S$: is the set of nodes visited by the *token* in BS
- $Leaves(\cdot)$: which takes as argument the current token location and returns the number of leaves in subtree rooted at that location
- $EulerLeaves(\cdot)$: which takes as argument the current token location and returns the number of leaves on the Euler tour starting from that location.

Essentially, Algorithm 2 expresses in a combinatorial way the conditions of ambiguity defined in Section 2.2

**Efficient implementation of** BS_Jumps**.** In this work, the main metric of consideration is the revealing time (or query complexity), in which the last agent was revealed. In this paragraph, we briefly discuss other metrics of efficiency of our protocol, such as the computational and communication complexity. The most time-consuming part of algorithm BS_Jumps is the token update. Because of the structure of binary tree, identifying ambiguous internal vertices or leaves that are in every $\phi$-compatible configuration could be done fast by each agent, in each step. In short, the agent simulates locally the Binary-Search-like process and evaluates whether the number of "remaining" leaves is large enough compared to the number of remaining active agents (the latter could be computed from the feedback $\phi$). If it is, it yields ambiguity, as the number of active leaves/agents in the subtree of the currently considered vertex could be $0$ or larger. As this local procedure is built on the top of Binary Search (simulated locally), it is efficient, in the sense that the number of local computational time in any agent is $O(k + k \log(n/k))$, which is $O(1)$ time amortized per round. Regarding the communication complexity of the protocol, it can be derived as a function of the revealing time of the protocol. In each round, each active agent communicates a single bit. The output of the ternary feedback can be encoded by $\log n$ bits. Let $revtime$ be the revealing time of the protocol. Thus, total communication complexity of a protocol is $(k + \log n) \cdot revtime$ bits. The total number of local steps during the protocol could be upper bounded by $\log n \cdot revtime$, because each step/round of algorithm BS_Jumps requires updating of the token, which is a logarithmic operation (see the Supplementary Materials, Section 3). Lastly, the memory needed in each agent is logarithmic, mainly due to the simple recursion in the main algorithm and the token update, and a constant number of variables. This is under assumption that any basic arithmetic operation on variables/values, represented by logarithmic numbers of bits, could be done in a single local step, while these numbers could be stored in a unit of memory.

**Theorem 3.1.** *Algorithm* BS_Jumps$(n, k)$ *is an $(n, k)$-AE, for any $2 \leq k \leq n$, and has revealing time $O(k \log(2n/k))$.*

For the technical proof of Theorem 3.1 we refer the reader to Section 6.

## 4 Equilibria for unknown size $k$

We first show in Section 4.1 that in the restricted feedback – with collision detection only – universal $n$-AE come at a high cost in performance measures. Intuitively, the high cost in that setting occurs

because the adversary has even more power, since it does not only select the configuration but also the size of it and the agents are agnostic to it. We then give a universal $(n, k')$-AE which is an $(n, k)$-CGT with maximum revealing time $O(n^{k-1})$, for any $2 \leq k \leq k' \leq n$. Here, $k$ is the actual number of active agents, unknown to the agents, while $k'$ is the known upper bound. In particular, the maximum revealing time is polynomial for constant-size configurations.

## 4.1 Equilibria characteristics and lower bounds

We start from proving that universal adversarial equilibria require (at least some) dense queries.

**Lemma 2.** *For any $3 \leq k' \leq n$, no algorithm with first query of size different from $n - 1$ is a universal $(n, k')$-AE.*

*Proof.* Suppose, to arrive with a contradiction, that there is an algorithm which is a universal $(n, k')$-AE, for some $3 \leq k' \leq n$, and has the first query of size different from $n - 1$. According to the definition of a universal $(n, k')$-AE, in order to get a contradiction we need to show that there is a deviation of some player $i$ such that for some $2 \leq k \leq k'$, for any configuration of size $k$ consistent with the deviation the revealing time of player $i$ does not increase while for at least one such configuration the revealing time decreases. In our arguments below, we will be showing that such contradiction actually occurs for $k = 2$ or $k = 3$.

The first query, call it $Q_1$, is fixed, as there is no feedback before it.

Consider first the extreme cases when $Q_1 = \emptyset$ or $Q_1 = [n]$.

*Case $Q_1 = \emptyset$:* player 0 could simply deviate by additionally adding itself to $Q_1$. This way, it improves in any configuration it is in, as it is always selected in the first round while before the deviation – in round bigger than 1. Hence, there is no worsening configuration (independently of what $k \geq 2$ is considered). Thus we get a contradiction with the definition of a universal $(n, k')$-AE.

*Case $Q_1 = [n]$:* first observe that the feedback is $\perp$ for any configuration of size $k' \geq 2$. Therefore, the second query, $Q_2$, is uniquely defined. Let $i$ be an arbitrary player not in $Q_2$, if $Q_2 \neq [n]$, or $i = 0$ otherwise (i.e., when $Q_2 = [n]$). Player $i$ deviates by removing itself from query $Q_1$ and adding to query $Q_2$ (if $i$ already belongs to $Q_2$, which is possible only if $Q_2 = [n]$, there is no need to adjust $Q_2$). Consider $k' = 2$. In any configuration of size 2 player $i$ is in, say $\{i, j\}$ for some $j \in [n] \setminus \{i\}$, before the deviation player $i$ is selected in round 2 at the earliest. However, after the deviation, player $i$ is selected always in round 2, as $j$ is selected in round 1. Hence, no configuration worsens for $i$ after deviation. Additionally, for configuration $\{i, j\}$ such that $j$ is in the original $Q_2$, before deviation $i$ is selected after round 2, which means that the deviation improves the revealing time of player $i$ in this configuration. Thus we get a contradiction with the definition of a universal $(n, k')$-AE.

Next, consider the case $1 \leq |Q_1| \leq n - 2$.

*Case $1 \leq |Q_1| \leq n - 2$:* Suppose that the feedback in round 1 is $v \in Q_1$, for some arbitrarily chosen $v \in Q_1$. Consider query $Q_2$ which comes after such feedback. There are two complementary sub-cases:

*Sub-case (a):* at least one element, say $i$, in $[n] \setminus Q_1$ is not in $Q_2$. Fix a deviation of element $i$ that after result $v$ in round 1, puts itself to query $Q_2$ (to which it originally does not belong). Consider $k = 2$. There is only one configuration of size 2 consistent with the deviation, containing element $v$ (the feedback in the first round) and element $i$ itself (note that $i \neq v$ since $v$ is in $Q_1$ while $i$ is not). In the configuration $\{v, i\}$, before deviation $i$ was selected in round bigger than 2 (as it does not belong to $Q_1$ and to $Q_2$), while after the deviation – in round 2. Hence we get a contradiction with the definition of a universal $(n, k')$-AE, since $\{v, i\}$ is the only configuration consistent with feedback $v$ in the first round.

*Sub-case (b):* all elements in $[n] \setminus Q_1$ are in $Q_2$. By the case assumption $|Q_1| \leq n - 2$, there are at least two such elements. Assume that the feedback in the second round is $\perp$, and consider the query $Q_3$ in round 3 of the algorithm (under feedback $v, \perp$ in the first two rounds). Consider $k = 3$.

First assume that there is at least one element, say $i$, which is in $Q_2 \cap ([n] \setminus Q_1)$ and is not in $Q_3$. Define a deviation of player $i$ which, after hearing feedback $v$ in the first round removes itself from query $Q_2$ but adds to the query in the third round of the execution, call it $Q'_3$. (Note that in such execution query $Q'_3$ could be different from $Q_3$, because it depends on the feedback in round 2 which

could be different from the original when player $i$ deviates.) For any configuration that is consistent with the history before the deviation and contains player $i$, say $\{v, j, i\}$ where $j \in Q_2 \setminus \{v, i\}$, before deviation player $i$ is selected after round 3 (as the feedback is $v, \perp$ in the first two rounds, as $i, j \in Q_2$, and $i$ is not in $Q_3$), while after the deviation the feedback in the first three rounds is $v, j, i$; hence the revealing time is improved. Such deviation for $k = 3$ yields contradiction with the initial assumption of being a universal $(n, k')$-AE.

Second, complementary to the previous assumption, consider a situation when all elements that are in $Q_2 \cap ([n] \setminus Q_1)$ are also in $Q_3$. Using the same deviation as in the previous argument, applied to any element $i \in Q_2 \cap ([n] \setminus Q_1) \cap Q_3$, and $k = 3$, we get that after the deviation element $i$ is selected in the third round for any configuration $\{v, j, i\}$ of three different element that is consistent with the history prior the deviation, while when $i$ has followed the algorithm the feedback in the first two rounds would be $v, \perp, \perp$ as $i, j \in Q_2 \cap Q_3$. This is again a contradiction with the algorithm being a universal $(n, k')$-AE. $\qquad\square$

Intuitively, by applying Lemma 2 to the first $n/2$ queries (assuming that the feedback is collision), we either get impossibility at some point or all these queries have to be of size $n - 1$. This intuition cannot be extended much beyond the first $n/2$ queries produced by an algorithm, because each query of size $n - 1$ somehow restricts the family of configurations of size 2 that we could use in the proof of Lemma 2.

**Theorem 4.1.** *Any algorithm that is an $(n, k')$-CGT and a universal $(n, k')$-AE has the maximum revealing time $\Omega(n)$, for any $k' \geq 3$. Moreover, the total number of elements in the queries in some executions are $\Omega(n^2)$.*

*Proof.* Fix any $3 \leq k \leq n$. We argue now that if an algorithm is a universal $(n, k)$-AE and an $(n, k)$-CGT, then it has maximum revealing time $\Omega(n)$. Suppose, to the contrary, that there is such an algorithm with maximum revealing time smaller than $n/2$. By Lemma 2, the first query of such algorithm has $n - 1$ elements. Assume that the feedback to each query of size $n-1$ is $\perp$, until a query of size smaller than $n - 1$ occurs, say at round $t$, for some $1 \leq t < n/2$. Note that such assumption is valid: there are configurations of size $k$ resulting in such feedback before round $t$, and round $t$ is well-defined because the algorithm has to ask queries of size smaller than $n - 1$ to have any feedback different than $\perp$, which is needed to be able to reveal configuration $K$ of size $k \geq 3$ (essentially, to distinguish between any two different configurations of size $k$). Note that any deviation of an active agent before round $t$ is worsening for some configuration of size 2: if an agent swaps 0 to 1, it could create collision while without switching it would be a round revealing the configuration. Similarly, if an agent swaps from 1 to 0 it may create situation of empty feedback, which does not reveal itself, while without swapping it could have revealed itself by being selected. Hence, in what follows, we consider only deviations that happen after round $t$.

We now extend the arguments from the proof of Lemma 2 to the considered scenario up to round $t$. First observe that cases $Q_t = \emptyset$, as well as sub-case (b) of case $1 \leq |Q_t| \leq n - 2$ – i.e., $([n] \setminus Q_t) \cap ([n] \setminus Q_{t+1}) \neq \emptyset$, where $Q_{t+1}$, which is the query after $Q_t$, gets feedback $v$ for some arbitrary $v \in Q_t$ – could be analyzed analogously as in the proof of Lemma 2, because the argument relies only on configurations of size $k = 3$ while all such configurations result in collisions in preceding queries of size $n - 1$ each. Note that this is consistent with the assumed feedback $\perp$ in those queries, as the intersections are of size at least 2.

It remains to analyze the analogous sub-case (a) of case $1 \leq |Q_t| \leq n - 2$ from the proof of Lemma 2: at least one element, say $i$, in $[n] \setminus Q_t$ is not in $Q_{t+1}$. Fix a deviation of agent $i$ that after feedback $v$ in round $t$, includes itself into query $Q_{t+1}$ (to which it originally does not belong). Consider $k = 2$. There is only one configuration of size 2 compatible with the deviation, containing agent $v$ (selected in round $t$) and agent $i$ itself (note that $i \neq v$ since $v$ is in $Q_t$ while $i$ is not). In the configuration $\{v, i\}$, before the deviation agent $i$ was revealed in round bigger than $t + 1$ (as it does not belong to $Q_t$ and to $Q_{t+1}$), while after the deviation – in round $t + 1$. Hence we get a contradiction with the definition of the universal $(n, k)$-AE, since $\{v, i\}$ is the only configuration consistent with feedback in the first $t$ rounds. $\qquad\square$

**Algorithm 3:** BB_GR$(n, k')$, pseudo-code for active agent $j$

---

**for** $x = 1, \ldots, k' - 1$ **do**

    // We execute blocks $B_x$, until first selection

    **for** $r = 1, 2, \ldots, \binom{n}{n-x}$ **do**

        $Q_r^x \leftarrow r$-th query in $\mathcal{F}_n^{n-x}$ (arbitrarily ordered)

        **if** $\mathsf{Feedback}(Q_r^x \cap K) = i$ *for some* $i \in K$ **then**

            $t^* \leftarrow r$

            $K \leftarrow \{i\} \cup ([n] \setminus Q_r^x)$  // Hidden set revealed

            **break**

**if** $\mathsf{Feedback}(Q_r^x \cap K) = j$ **then**

    **for** $r = t^* + 1, \ldots, \binom{n}{n-x}$ **do**

        // We continue in block $B_x$ but with grim trigger

        $Q_r^x \leftarrow \{j\}$  // Persistent inclusions

    **if** $x < k' - 1$ **then**

        **for** $r = 1, 2, \ldots, \binom{n}{n-(x+1)}$ **do**

            // We run one more block $B_{x+1}$ with grim trigger when $x < k' - 1$

            $Q_r^{x+1} \leftarrow \{j\}$  // Persistent inclusions

        $Q_1^{x+2} \leftarrow \{j\}$  // Last persistent inclusion when $x < k' - 1$

    **else**

        $Q_1^{x+1} \leftarrow \{j\}$  // Last persistent inclusion when $x = k' - 1$

**Output** $K$

---

### 4.2 Equilibrium algorithm for unknown $k$

We now present the algorithm called *Block Builder with Grim Trigger* (or BB_GR for short), see Algorithm 3 for a pseudo-code. The structure of algorithm BB_GR$(n, k')$, where $k' \leq n$ is the known upper bound on $k$, consists of some $x' \leq k'$ subsequent blocks of queries. In each block of queries $B_x$, for an integer $x$ such that $1 \leq x < x'$, there are $y_x = \binom{n}{n-x} = \binom{n}{x}$ queries, while the last block, $x'$, contains only one query. To simplify the notation for the sake of this algorithm, but with some abuse of the notation used in previous sections, herewe denote the $r$th query of block $B_x$ as $Q_r^x$. Until the first selection of an element, we define a query $Q_r^x$ as the $r$-th set from the family $\mathcal{F}_n^{n-x}$ (where the order of these sets could be fixed arbitrarily by the algorithm). When the first successful selection happens, say for some query $Q_r^{x^*}$ in block $x^*$ such that $x^* = x' - 2$ (or $x^* = k' - 1$ in some cases when $x' = k'$, see the second part of the pseudo-code when the value of $x$ has been set to $k' - 1$), the hidden set $K$ is revealed as

$$K = \left\{ \mathsf{Feedback}(Q_r^{x^*} \cap K) \right\} \cup \left( [n] \setminus Q_r^{x^*} \right) .$$

After the first selection, the selected agent with $id = \mathsf{Feedback}(Q_r^{x^*} \cap K)$ includes itself persistently in the following queries up to query $Q_1^{x'}$, which is the first query of the last block $B_{x'}$ executed by the algorithm in the current game. Then, the algorithm stops following the block structure and terminates.

In the analysis of BB_GR$(n, k')$ below we show that the block number of the revealing time, $x^*$, satisfies $x^* = k - 1$, where $k$ is the actual size of a configuration. Consequently, the number of blocks in the game execution is $x' = k + 1$ or $x' = k$, where the latter occurs when $k = k'$ (see the second part of the algorithm's pseudo-code, with these two cases).

**Lemma 3.** BB_GR *is an $(n, k)$-CGT, for any $2 \leq k \leq k'$, with maximum revealing time $O(n^{k-1})$.*

*Proof.* Consider that the adversary chooses $k \in [2, k']$. In this case, the first successful selection will happen in the block $B_{k-1}$ and let us also name the respective query $Q_r^{k-1}$. The first selection will happen in the block $B_{k-1}$ because for every round in all the previous blocks of queries $B_x$ where $x < k - 1$, at least two active agents were included in the queries, since the number of agents who were not asked was $n - (n - x) = x < k - 1$. The hidden set $K$ comprises of the id of the selected agent which is given by $\mathsf{Feedback}(Q_r^x \cap K)$ plus the remaining $k - 1$ agents that were not included

in this query. The remaining $k-1$ agents are $[n] \setminus Q_r^x$. Thus, the hidden set $K$ is revealed. Let us now denote the time of the first selection as $t^*$ (the time when query $Q_r^{k-1}$ happened). It holds that $t^* = \sum_{1 \leq x < k-1} \binom{n}{x} + r$. Since $\binom{n}{x} \in O(n^x)$, this implies that $t^*$ is $O(n^{k-1})$ for any $k \leq k' \leq n$. $\square$

**Lemma 4.** `BB_GR` *is a* universal $(n, k')$-*AE.*

*Proof.* Now, we will prove that deviations are not profitable. As noted before, it is critical to keep in mind that the characteristic property of this algorithm is when the first selection happens, subsequently, the whole hidden set is revealed at the same round. Consider that adversary selects $k$. First, we argue that deviations from 0 to 1 (i.e., when an agent was excluded from the query and it included itself) by agent $i$ are not profitable. It is easy to see that such deviations do not make any difference in the first $k-2$ blocks of queries since in every round there will be at least two active agents included in the queries and hence the feedback will be *collision* independently of what $i$ does. In block $B_{k-1}$, depending on the configuration selected by the adversary, the only rounds when agent $i$ can be selected are the ones allocated by the algorithm, in which $i$ is the only active agent in the query. In the other queries of block $B_{k-1}$, there will be at least one more active agent included since $|Q_r^{k-1}| = n - (k-1)$. We consider now deviations from 1 to 0 (i.e., when the agent was included in the query and it excluded itself) by agent $i$. Following the same reasoning, deviations by agent $i$ do not influence the feedback channel up until block $k-2$, since for each round there will be at least two active agents included in the query and thus the output of the feedback will be *collision*. Suppose now that agent $i$ deviates in block $B_{k-2}$ by not including itself in a query $Q_r^{k-2}$ where it was initially included by the algorithm. And more specifically we consider a configuration where agent $j$ is also active, that is $j \in K$, and that out of $|Q_r^{k-2}| = n - (k-2)$ included in the query only $i, j \in K$. Thus, the deviation of $i$ will lead agent $j$ to be be selected in query $Q_r^{k-2}$. Since $j$ follows the algorithm, it will include itself to the following queries up to query $Q_1^k$ and hence this will render agent $i$ unable to reveal itself up until that round. Thus this deviation is not profitable for $i$ because if it didn't deviate it would have been revealed in some round of block $B_{k-1}$, depending on the configuration chosen by the adversary. The same reasoning also holds for deviations of agent $i$ from 1 to 0 in block $B_{k-1}$. Thus we conclude that deviations are not profitable. $\square$

**Theorem 4.2.** `BB_GR` *is universal* $(n, k')$-*AE and* $(n, k)$-*CGT with maximum revealing time* $O(n^{k-1})$ *for any number of active agents $k$ such that $k \leq k' \leq n$.*

*Proof.* The maximum revealing time follows from Lemma 3. Being a universal $(n, k')$-AE follows from Lemma 4. $\square$

## 5 Extensions and applications

### Contention Resolution Games

In the *Contention Resolution Game (CR Game)*, there are $n$ selfish agents (players), each having a single packet to be broadcast on a single shared channel. We will use the term agent and player interchangeably. Each agent has a unique *name (id)* which is an integer in the range $[n] = \{0, 1, \ldots, n-1\}$, and they are modeled as elements $[n]$ in the CGT problem. The communication is in synchronous rounds, also called steps, interchangeably.

**A shared channel.** An agent *transmitting* in the Contention Resolution problem in round $t$ corresponds to the agent being included in the query $Q_t$ within the CGT model. The revealing time of an agent in the CGT game where the agent is *selected* corresponds to a successful transmission in the CR game. In the CR game, if only one agent transmits a message in a round then the transmission is successful. If two or more agents transmit their packets at the same round then there is a collision on the channel and none of them is successful. Agents attached to the channel receive feedback in each round. In the model *without collision detection*, we assume that each player hears feedback from the channel in each round, which is: silence (if nobody or at least two players transmit at that round) or busy channel (if exactly one player transmits). In the model *with collision detection*, each player receives ternary feedback in each round, which is: silence (in nobody transmits), collision (if at least two players transmit at that round), or busy channel (if exactly one player transmits).

**Contention resolution (CR) games.** A distributed communication algorithm executed by an agent serves as its strategy. We consider only deterministic algorithms. An algorithm determines for each round if the agent transmits or pauses, or possibly halts and exits. An algorithm is *non-adaptive* if the sequence of attempts to transmit and pauses for each individual agent is determined in advance and encoded as a sequence of zeros and ones. Such non-adaptive algorithms are studied in paper Chionas et al. [2023].

## 5.1  CR Games: Feedback with collision detection

An *adaptive* algorithm, studied here, can determine in each round whether the player transmits, pauses or halts and exists, based on the complete history of the channel feedback up to the current round. There is a natural translation of all notions and algorithms defined for CGT to the corresponding notions and algorithms for the CR games. In this way, all our results in theorems for CGT translate naturally to CR games.

## 5.2  CR Games: Restricted feedback without collision detection

We present in Subsections 5.2 and 5.3 our extensions of these result from Subsection 5.1 in the CGT model to CR games with restricted feedback. We assume in Subsections 5.2 and 5.3 that each station hears feedback from the channel in each round, which is: silence (if nobody or at least two players transmit) or a transmission (if exactly one player transmits). We slightly modify the feedback history here, where $\phi[t] = 0$ will denote that there was silence in round $t$, and $\phi[t] = 1$ – denoting a transmission in round $t$ (in this case 1 does not correspond to the agent's id, as in this case any of the agents in $[n]$ successfully transmits).

Chionas et al. [2023] introduced a non-adaptive modification of Round-Robin principle, called Persistent_RR, in which a player $i$ who fails to successfully transmit in its dedicated slot $i + 1$, starts grim-trigger sequence until slot $n$. Surprisingly, this is not an equilibrium if adaptive algorithms are considered.

---

**Algorithm 4:** Persistent_RR_Jump$(n)$, player $i$

---

$succ = 0$
**for** $t \in [n]$ **do**
    **if** $t = i$ **or** $succ = k - 1$ **then**
        $transmit$(packet)  *(switch-off upon ack)*
    **if** $t > i$ **then**
        $transmit$(noise)
    **if** $\phi[t] = 1$ **then**
        $succ$ ++

---

Consider a strategy when each player with id $i$ transmits in slot $i + 1$ or in the next slot after the slot when the $(k - 1)$'st successful transmission was heard, whatever comes first. Consider all configurations in which $i > k$ is the player with largest id; they exist since $n > k$. For some of these configurations, there are less than $k - 1$ successful transmissions before slot $i + 1$, the player $i$ transmits in its slot $i + 1$ complying with Persistent_RR and does not change its Persistent_RR-based latency. However, there is a configuration $K$ such that player $i - 1$ does not belong to it (again, since $i > k$ and $i$ is the largest id in the configuration), and in such case $i$ transmits successfully before slot $i + 1$ without causing any collision and thus not triggering the persistent transmissions.

**Lemma 5.** *For known $k \geq 2$ and $n > k$, Persistent_RR$(n)$ is not $(n, k)$-AE.*

*Proof.* Consider a strategy when each player with id $i$ transmits in slot $i + 1$ or in the next slot after the slot when the $(k - 1)$'st successful transmission was heard, whatever comes first. Consider all configurations in which $i > k$ is the player with largest id; they exist since $n > k$. For some of these configurations, there are less than $k - 1$ successful transmissions before slot $i + 1$, the player $i$ transmits in its slot $i + 1$ complying with Persistent_RR and does not change its Persistent_RR-based latency. However, there is a configuration $K$ such that player $i - 1$ does not belong to it (again, since $i > k$ and $i$ is the largest id in the configuration), and in such case $i$ transmits successfully before slot $i + 1$ without causing any collision and thus not triggering the persistent transmissions. $\square$

Consider the following adaptive modification to Persistent_RR in the model without collision detection, motivated by the proof of Lemma 5: if some player just heard $(k-1)$-st successful transmission, it transmits in the very next round. We call this new algorithm Persistent_RR_Jump, see the pseudocode of Algorithm 4 in the Appendix.

Before we prove that Persistent_RR_Jump is an $(n, k)$-AE, we will show how to model any adaptive (strategy) algorithm $\sigma$ in the setting without collision detection. Let $\phi[1, \ldots, t-1]$ be a given feedback history, for any $t \geq 1$. If $t = 1$, then the history $\phi[1, \ldots, t-1]$ is empty and the algorithm $\sigma$ takes a deterministic decision of whether to transmit or not in the first round, without any prior knowledge. If $t \geq 2$, then the algorithm $\sigma$ takes a deterministic decision of whether to transmit or not in round $t$, based on the history $\phi[1, \ldots, t-1]$. In particular the algorithm $\sigma$ knows how many among the $k$ players successfully transmitted by time $t-1$.

**Theorem 5.1.** *Algorithm Persistent_RR_Jump is an adversarial equilibrium, $(n, k)$-AE, with maximum latency $n$.*

*Proof.* The maximum latency $n$ follows from the case when both id's $n-2$ and $n-1$ are present in the configuration set. It remains to prove that the algorithm is an adversarial equilibrium. Recall that in Persistent_RR if player $i \in [n]$, before deviation, transmits for the first time at round $i+1$, and if unsuccessful, it transmits persistently from round $i+2$ onward until round $n$.

Let us fix a player $i \in [n]$ who follows the strategy $s_i = (0, 0, \ldots, 0, 1, 1, \ldots, 1)$, where the first 1 is in slot $i+1$ and strategy $s_i$ is augmented by the above modification. That is, player $i$ follows the Persistent_RR_Jump strategy $s_i$. Let strategy (algorithm) $s_i'$ be any deviation of player $i$ from its strategy $s_i$; that is, algorithm $s_i'$ is *any* deterministic algorithm modeled as defined above. We will prove that (excluding some obvious cases), there always exists a worsening configuration $K \in \mathcal{F}_n^k$, $i \in K$, for player $i$ under the deviation $s_i'$.

Let us choose any feedback history $\phi \in \{0, 1\}^*$. Suppose that $t \in \{1, \ldots, n\}$ is the round of the first deviation, that is, the first time $t$, where the algorithm $s_i'$ took a different decision from that of algorithm $s_i$. Suppose that this happened after feedback history $\phi[1, \ldots, t-1]$ and under configuration $K \in \mathcal{F}_n^k$, that was $\phi$-compatible with algorithm $s_i'$.

We observe first that if $t \geq i+2$, then player $i$ cannot have any improvement in its transmission time, therefore for such deviation there is no improving configuration. Assume otherwise, that $t \leq i+1$. If $t = i+1$ and player $i$ deviated from $s_i[t] = 1$ to $s_i'[t] = 0$, then configuration $K$ itself is worsening for player $i$.

Suppose now that $t \leq i$. The first case is when the algorithm $s_i$ told player $i$ to not transmit at time $t$ after history $\phi[1, \ldots, t-1]$, and the deviator algorithm $s_i'$ transmitted at time $t$. This means that under configuration $K$, by time $t-1$ inclusive, at most $k-2$ players from $K$ successfully transmitted. Let $K' \subset K$, $|K'| \leq k-2$ be the set of those players who successfully transmitted by time $t-1$. Notice that $i \notin K'$ because time $t$ was the first deviation of player $i$ from strategy $s_i$.

We now define a new configuration $K'' = K' \cup \{i, t-1\}$, which contains player $t-1$ such that $s_{t-1} = (0, 0, \ldots, 0, 1, 1, \ldots, 1)$, where the first 1 is in slot $t$ and strategy $s_{t-1}$ is augmented by the above modification (Persistent_RR_Jump). Notice that $t-1 \notin K'$, because the first transmission time of player $t-1$ could only be $t$, as we know that until time $t-1$ at most $k-2$ players successfully transmitted. Moreover if $|K''| < k$, we add to $K''$ any players from $K \setminus K''$ to make its size exactly $k$; this change clearly will not affect the claim that until time $t-1$ at most $k-2$ players successfully transmitted under this new configuration $K''$. Under configuration $K''$, player $t-1$ will block player $i$'s deviation at time $t$, and because player $t-1$ is unsuccessful at time $t$, it will start persistently transmitting from time $t+1$ onward, completely blocking player $i$ who plays its deviation $s_i'$, under configuration $K''$. Thus, configuration $K''$ is worsening for player $i$.

The second case is when the algorithm $s_i$ tells player $i$ to transmit at time $t$ after history $\phi[1, \ldots, t-1]$, and the deviator algorithm $s_i'$ does not transmit at time $t$. This means that under configuration $K$, by time $t-1$ inclusive, exactly $k-1$ players from $K$ successfully transmitted. Because player $i$ could have safely transmitted in time $t$, and has chosen not to do so in $s_i'$, configuration $K$ is worsening for player $i$. This finishes the proof of the theorem. $\square$

### 5.3 CR Games: Restricted feedback without revealing id

We will show here how to modify our algorithms so that in addition to still being adversarial equilibria, they let the players to learn the ids of all other active players, while achieving the same maximum latency as before. All the players in a configuration $K$ chosen by the adversary, listen to the channel (even after they successfully transmitted, due to the adaptivity) and based on the channel feedback they will be able to learn other players ids, as we will show below.

**Theorem 5.2.** *Algorithm Persistent_RR_Jump can be modified such that it is an adversarial equilibrium, $(n, k)$-AE, with maximum latency $n$, and it allows for learning of id's of the active players.*

*Proof.* We modify the algorithm Persistent_RR_Jump for player $i \in [n]$ as follows: if player $i$ hears that the $(k-1)$-st player successfully transmitted at time $t$, then $i$ transmits at time $t + 1$ and then if $t + 1 < i + 1$, then $i$ persistently transmits in slots $t + 2, t + 3, \ldots, i + 1$, and switches off after that.

Notice that in the original (and modified) Persistent_RR_Jump the first $k - 1$ players transmit (without deviations) at time that is their id+1, thus allowing other players to learn their id's. The last, $k$-th, player might transmit in the modified algorithm Persistent_RR_Jump before its original slot id+1, but then it will keep transmitting until time id+1 inclusively. Thus, its last transmission will let the other players learn its id.

Obviously, this modification does not increase the maximum latency beyond $n$. We will argue now that it is still an $(n, k)$-AE. Let us fix any player $i \in [n]$, who follows the strategy $s_i$ according to the modified algorithm Persistent_RR_Jump. Let us also denote by $s_i'$ any deterministic algorithm that is any deviation of player $i$ from strategy $s_i$.

Suppose that $t \in \{1, \ldots, n\}$ is the round of the first deviation, that is, the first time $t$, where the algorithm $s_i'$ took a different decision from that of algorithm $s_i$ for some configuration $K$. Suppose that this happened after feedback history $\phi[1, \ldots, t-1]$ and under configuration $K \in \mathcal{F}_n^k, i \in K$, that was $\phi$-compatible with algorithm $s_i'$ (as then it has not deviated from the original $s_i$).

The first case to consider is that at round $t$ there are still at least two active players from $K$, including player $i$. Because up to round $t$ the original and modified algorithm Persistent_RR_Jump are identical, such deviation $s_i'$ is not profitable for player $i$ by the proof of Theorem 5.1.

The second case to consider is that at round $t$ there is only one active player from $K$, i.e., player $i$. Because $t$ is the first round where player $i$ deviated, there exists a round $t' < t$ at which the $(k-1)$-st player successfully transmitted. Without deviation, player $i$ should transmit at time $t' + 1$, and if $i$ has not done so, configuration $K$ is worsening for $i$. No deviation of player $i$ after time $t' + 1$ can be profitable for that player as well. □

In case of restricted feedback – only either silence or 1 in case of success (no id's are revealed), we need a new mechanism for learning agents' id's. The mechanism is as follows. A player $i$ transmits a sequence of $\log(n)$ 0's (silence) and 1's which code its id, i.e., number $i \in [n]$ in binary. We use this mechanism in the proof of the theorem below.

**Theorem 5.3.** *Algorithm $\texttt{BS\_Jumps}(n, k)$ can be modified in such a way that it is an $(n, k)$-AE, for any $2 \le k \le n$, has maximum latency $O(k \log(2n/k))$, and it allows for learning of id's of the active players.*

*Proof.* We will use the position of the token in the original algorithm $\texttt{BS\_Jumps}(n, k)$ to learn the id's of the players. The position of the token $v$ encodes the following number in binary, which we call the *token code*: following the path in the binary tree starting from the root to $v$, any time we go left to a child, we append at the end of the code bit 0, and whenever we go right, we append the bit 1. Clearly, if token $v$ is at a leaf, say $i \in [n]$, of the binary tree, the token code is equal to $i$, the id of that player. Therefore, whenever in the original algorithm $\texttt{BS\_Jumps}(n, k)$, a leaf $i$ successfully transmits and the token is at that leaf, all other players can learn the id of player $i$.

We will show next how to modify algorithm $\texttt{BS\_Jumps}(n, k)$ in case when a player $i$ successfully transmits but the token is in an internal vertex $v$. By correctness of $\texttt{BS\_Jumps}$, $v$ is an ancestor of $i$ in the tree. By similar argument as above, the token code is a correct prefix (of length $y$, corresponding to the depth of $v$) of the binary id of player $i$. Instead of continuing to the next step (and token

location) of the `BS_Jumps`, all players freeze that algorithm for $x$ rounds, where $x$ is the length of a path from $v$ to leaf $i$. During that time, only player $i$ could transmit, and it does it according to the digits of its id beyond the prefix consistent with the token code of $v$ – we call it *token descend*. More precisely, in the $j$-th round of the frozen period (of $x$ rounds in total), if player $i$ has 1 in position $y + j$ of its id, it transmits, otherwise it stays silent. All players receive feedback 1 on such transmissions and feedback 0 on silences, which allow them to uniquely decode the remaining $x$ bits (to be concatenated with the token code, in order to get the full id of player $i$). However, if a collision is heard at any point of a token descend, the player $i$ who is actively executing the token descend starts the grim-trigger sequence of $4n$ consecutive transmissions (and switch off after it). This is to discourage any deviator from interrupting the learning process. After the token descend finishes successfully, i.e., a whole id of a leaf is discovered and grim-trigger has not started (no collisions during the token descend), the algorithm `BS_Jumps` is resumed by all active players.

Observe that the maximum latency remains $O(k + k \log(n/k))$, as `BS_Jumps` enhanced by the token descend described above is still not slower than the original Binary Search; hence Lemmas 8 and 9 could be used.

Finally, we need to argue what happens if a deviation occurs during token descend. If the first deviation occurs during token descend (recall that all other cases were already covered by Lemmas 10 and 12), we apply the same construction of configuration as done recursively in the proof of Lemma 12, which leads to grim-trigger sequence scheduled by the player who is doing the token descend. Clearly, it would be a worsening configuration. □

## 5.4 Applications of the framework to blockchain related problems

Our algorithm `BS_Jumps` resolves contention in a multi access single link channel and the algorithm is also adversarial equilibrium. Such an algorithm can be applied in a multi agent setting like blockchain in order to achieve consensus. Similarly with the context of a single link channel, in blockchain, only one block proposer at a time can extend the chain. If more than one agent creates new blocks, there would be a collision (a.k.a., fork of the chain). There are different mechanisms to achieve consensus in blockchain but the most well-known are Proof-of-Work (PoW) Nakamoto [2009] and Proof-of-Stake (PoS) Nguyen et al. [2019], such as Ouroboros, Algorand and Tendermint. Our algorithm can be considered as a modification of PoS. We can interpret our algorithm as achieving consensus for the the next $k$ blocks. Each block producer wants its block to be included in the chain as soon as possible and receive the respective block reward. We assume that the protocol distributes to $n$ block proposers different ids randomly. Then, for the next $k$ blocks the protocol selects a configuration of $k$ block proposers. Those $k$ agents will propose blocks based on the algorithm `BS_Jumps`$(n, k)$, which takes as input their ids. Each block producer is disincetivized to deviate from the given strategy since we have proven that by deviating there exists another configuration in which the agent will not be able to propose a block.

# 6 Analysis of Algorithm `BS_Jumps` – Proof of Theorem 3.1

We start by proving some technical invariants regarding properties of `BS_Jumps`$(n, k)$ before any deviation. Let $\phi$ be a feedback history, and for the sake of smooth argument, assume that $\alpha_i[0]$ is the root of the tree. The invariant for time step $t$ is as follows:

**Assumption:** Assuming a fixed feedback history $\phi$ up to $t - 1$ and assuming that there was no deviation by any agent by a step $t \geq 1$,

***Inv$_t$ (i):*** for any $t' < t$ such that $\phi[t'] \in [n]$ and for any configuration $K \in \mathcal{K}_\phi$, $|K \cap Q_{\alpha_i[t']}| = 1$, for any $i \in K$;

***Inv$_t$ (ii):*** for any $t' < t'' < t$ such that $\phi[t'] = i$, $\phi[t''] = j$, queries $Q_{\alpha_i[t']}$ and $Q_{\alpha_j[t'']}$ are disjoint, for any $i, j \in K$, $i \neq j$;

***Inv$_t$ (iii):*** let $x$ be the number of steps $t' < t$ with $\phi[t'] \in [n]$, then for any agent $i$ active in step $t$:

- if $\phi[t - 1] = \emptyset$ or $\phi[t - 1] \in [n]$, then for any subset $A$ of $k - x$ leaves located after the last leaf below $\alpha_i[t - 1]$ in the Eulerian tour, there is a configuration $K \in \mathcal{K}_\phi$ such that $A \subseteq K$;

- if $\phi[t-1] = \perp$ then for any subset $A$ of $k-x$ leaves located after the vertex $\alpha_i[t-1]$ in the Eulerian tour such that $|A \cap Q_{\alpha_i[t-1]}| \geq 2$, there is a configuration $K \in \mathcal{K}_\phi$ such that $A \subseteq K$;

**$Inv_t$ (iv):** for any two agents $i,j$ active in a step $t' \leq t$, their token locations in step $t'$ are the same, i.e., $\alpha_i[t'] = \alpha_j[t']$;

**$Inv_t$ (v):** there is a configuration $K \in \mathcal{K}_\phi$ such that $K \cap Q_{\alpha_i[t]}$ is a singleton, for any $i \in K$.

Note that $Inv_t$ (iv) ensures that token locations, and thus subsequently asked queries, are consistent for all active agents. $Inv_t$ (i), (ii) and (iii), on the other hand, characterize any $\phi$-compatible configuration: identify unique subtree of single agents/leaves prior to the current token location and any configuration of the remaining agents after the token location. Finally, $Inv_t$ (v) says that for an ambiguous query $Q_v$ there is a $\phi$-compatible configuration $K$ having exactly one element in $Q_v$, i.e., ambiguous queries cannot have only configurations resulting in feedback $\emptyset$ or $\perp$. Now we prove $Inv_t$ formally.

**Lemma 6.** *For any feedback sequence $\phi$ fixed up to some step $t-1$, and assuming there were no deviations of any agent by step $t$, the Invariant $Inv_t$ for $t$ holds.*

*Proof.* The proof is by induction on $t$. In the beginning ($t = 1$) all the five properties clearly hold, by the initialization of the algorithm.

Suppose that the Invariant holds up to some time $t-1 \geq 1$, and we will prove it for $t$.

**Proof of $Inv_t$ (i).** By inductive assumption for $Inv_{t-1}$ (i), the statement of $Inv_t$ (i) holds for any $t' < t-1$. By $Inv_{t-1}$ (iv), the token locations were consistent in each step $t' \leq t-1$ across active players in a configuration $K \in \mathcal{K}_\phi$. Hence, the feedback $\phi[t']$ was correctly answered based on the size of intersection $K \cap Q_{\alpha_i[t']}$, where $i$ is an arbitrarily chosen player active in step $t' \leq t-1$. Hence, since $\phi[t-1] \in [n]$ then indeed $|K \cap Q_{\alpha_i[t-1]}| = 1$ and $Inv_t$ (i) follows also for $t' = t-1$.

**Proof of $Inv_t$ (ii).** The corresponding queries $Q_{\alpha_i[t']}, Q_{\alpha_i[t'']}$ have to be disjoint, because the token after receiving feedback in $[n]$ jumps to some next vertices of the Eulerian tour, and they have different sets of leaves below them. (Here, similarly as in the proof of $Inv_t$ (i), we also use $Inv_{t-1}$ (iv) about consistency of token locations across active processes in a step smaller equal to $t-1$.) Hence, $Inv_t$ (ii) holds.

**Proof of $Inv_t$ (iii).** Note that if $\phi[t-1] \in \{\emptyset, \perp\}$ then the value of $x$ is the same as in the inductive assumption for $t-1$. If $\phi[t-1] = \emptyset$ then also no configuration $K \in \mathcal{K}_\phi$ has any leaf in $Q_{\alpha_i[t-1]}$, thus all other $k-x$ leaves are after the last leaf below $\alpha_i[t-1]$ on the Eulerian tour. If $\phi[t-1] = \perp$ then at least two leaves under $Q_{\alpha_i[t-1]}$ must belong to $A$ and all $k-x$ elements must be after $\alpha_i[t-1]$ on the Eulerian tour. Finally, in case $\phi[t-1] \in [n]$, we have $K \cap Q_{\alpha_i[t-1]}$ is a singleton. Hence, the value $x$ is bigger by 1 compared to $Inv_{t-1}$, and any set $A$ of $|A| = k-x$ leaves on the right of the last leaf below $\alpha_i[t-1]$ is in some $\phi$-compatible configuration $K$. Formally, we take any $A'$ satisfying $Inv_{t-1}$, of size $k-x+1$, which contains a single intersection with $Q_{\alpha_i[t-1]}$, and observe that $A = A' \setminus Q_{\alpha_i[t-1]}$ could be any set of $k-x$ leaves on the right of the last leaf below $\alpha_i[t-1]$. And since $A'$ was a subset of some $\phi$-compatible configuration $K$ (until step $t-2$), $A$ is a subset of the same configuration $K$ and $K$ is also compatible until step $t-1$ (as, by $Inv_{t-1}$ (ii) and the fact that $A' \setminus A \subseteq K$ is a singleton, $K \setminus A$ has $x$ elements being singletons in queries $Q_v$ corresponding to token locations $\alpha[t'] = v$ such that $\phi[t'] \in [n]$, which also includes $t' = t-1$).

**Proof of $Inv_t$ (iv).** It is enough to prove that if for an internal vertex $v$ in the Eulerian tours (which are the same for all active agents, as they start from the same node, by inductive assumption) the query $Q_v$ is ambiguous for an active agent $i$, then it is also ambiguous for any other active agent $j$. Being ambiguous for agent $i$ means there are $\phi$-compatible configurations $K_1, K_2$ containing agent $i$ that give different feedbacks on query $Q_v$. Consider another active agent $j \neq i$. It is enough to consider the following three cases (others are symmetric and obtained by swapping roles of $K_1, K_2$). In all of them we consider only cases when $j$ does not belong to some of $K_1, K_2$, as otherwise those configurations would also prove ambiguity of vertex $v$ for agent $j$.

Case 1: $K_1 \cap Q_v = \emptyset$ and $K_2 \cap Q_v = \{j'\}$, for some $j' \in [n]$.

Note that $j' \neq i$ since $j' \in Q_v$, $i \in K_1$ and $K_1 \cap Q_v = \emptyset$. If $j' \neq j$ then we define $K_1' = (K_1 \setminus \{i\}) \cup \{j\}$ and $K_2' = (K_2 \setminus \{i\}) \cup \{j\}$. Observe that if $j \in Q_v$ then $K_1' \cap Q_v = \{j\}$ and $K_2' \cap Q_v = \{j, j'\}$, thus $K_1', K_2'$ resulting in two different feedbacks on query $Q_v$. Similarly, if

$j \notin Q_v$ then $K_1' \cap Q_v = \emptyset$ and $K_2' \cap Q_v = \{j'\}$, thus again $K_1', K_2'$ resulting in two different feedbacks on query $Q_v$. Hence, $Q_v$ is ambiguous for agent $j$ as well. It remains to consider sub-case $j' = j$. Since $v$ is an internal vertex, $Q_v$ contains at least two elements. Let $j'' \neq j'$ be another element in $Q_v$. Consider configurations $K_1' = (K_1 \setminus \{i\}) \cup \{j\}$ and $K_2' = (K_2 \setminus \{i\}) \cup \{j''\}$. They both contain $j = j'$ and $K_1' \cap Q_v = \{j\}$ while $K_2' \cap Q_v = \{j, j''\}$, hence the feedback is different for agent $j$ and $v$ is thus ambiguous.

Case 2: $|K_1 \cap Q_v| \geq 2$ and $K_2 \cap Q_v = \{j'\}$, for some $j' \in [n]$.

We define $K_1' = (K_1 \setminus \{i\}) \cup \{j\}$ and $K_2' = (K_2 \setminus \{i\}) \cup \{j\}$. Assume first that $i = j'$. If $j \notin Q_v$ then both $K_1', K_2'$ contain agent $j$ and result in feedback in $[n] \cup \{\bot\}$ in case of $K_1'$ and feedback $\emptyset$ in case of $K_2'$, thus $v$ is ambiguous also for agent $j$. If $j \in Q_v$ then the feedback for $K_1'$ and $K_2'$ would be $\bot$ and in $[n]$, respectively, which again proves $v$ being ambiguous for agent $j$. It remains to consider case $i \neq j'$. If $j' \neq j$ then we re-define $K_2' = (K_2 \setminus \{j'\}) \cup \{j\}$. Now, if $j \in Q_v$ we have $|K_1' \cap Q_v| \geq 2$ and $K_2' \cap Q_v = \{j\}$, thus $K_1', K_2'$ resulting in two different feedbacks on query $Q_v$.

Similarly, if $j \notin Q_v$ then $|K_1' \cap Q_v| \geq 1$ and $K_2' \cap Q_v = \emptyset$, thus again $K_1', K_2'$ resulting in two different feedbacks on query $Q_v$. Hence, $Q_v$ is ambiguous for agent $j$ as well. It remains to consider sub-case $j' = j$. Since it implies that $j \in K_2$, we have $j \notin K_1$. Hence, we get feedback $\bot$ on configuration $K_1'$ and feedback in $[n]$ on $K_2$, both configurations containing agent $j$. Thus, $v$ is also ambiguous for agent $j$.

Case 3: $|K_1 \cap Q_v| \geq 2$ and $K_2 \cap Q_v = \emptyset$.

In this case, we could create a configuration $K_2' = (K_2 \setminus \{j''\}) \cup \{j'\}$ for any $j' \in Q_v$ and any $j'' \in K_2$ being after leaves in $Q_v$ in the Eulerian tour. This way we consider $K_1$ and $K_2'$ as in Case 2. Note that $j'$ obviously exists as $Q_v$ is non-empty, while $j''$ exists because of $\mathrm{Inv}_t$ (iii) (note here that for a given $t$, we prove invariants in the order from (i) to (v)).

***Proof of $\mathrm{Inv}_t$ (v).*** By the token update rule, $v$ is either a leaf belonging to any configuration in $\mathcal{K}_\phi|_i$, for some active agent $i$ – which automatically confirms existence of the desired configuration – or is an internal ambiguous vertex. In the latter, the argument is exactly the same as in Case 3: if we had ambiguity between feedbacks $\emptyset$ and $\bot$, we could construct another $\phi$-compatible execution with feedback in $[n]$.

This completes the proof of Lemma 6. $\qquad\square$

The next lemma shows that, without any deviation, the token locations are the same across all active agents.

**Lemma 7.** *If there are at least two active agents in the beginning of step $t$ of an execution of* BS_Jumps$(n, k)$ *and there was no deviation by step $t - 1$, then for any two such agents $i, j$ their token locations in step $t$ are the same, i.e., $\alpha_i[t] = \alpha_j[t]$.*

*Proof.* The proof is by induction on step $t$. For step 1 every agent sets its token location to the left-most child on the root, hence they are the same. Suppose the lemma holds up to step $t - 1$; we now prove it for step $t$.

It is enough to prove that if for an internal vertex $v$ in the Eulerian tours (which are the same for all active players, as they start from the same node, by inductive assumption) the query $Q_v$ is ambiguous for an active agent $i$ then it is also ambiguous for any other active agent $j$. We have to prove here that the locations of the token $\alpha_i[t]$ and $\alpha_j[t]$ are the same at the beginning ot step $t$. But by our assumption, there is no deviation of any player at the beginning of step $t$. We can therefore prove the claim by exactly following the proof of $\mathrm{Inv}_t$ (iv). $\qquad\square$

The next result compares the tokens' locations in parallel executions of algorithms BS_Jumps$(n, k)$ and BS (Binary Search), both on the same configuration.

**Lemma 8.** *Consider a configuration of agents $K$, and their concurrent executions of algorithms* BS *and* BS_Jumps$(n, k)$ *without deviations. Then, in any step $t$:*

*(i) if an agent is active in both executions then its token in the latter is not behind the token in the former, with respect to the left-to-right Eulerian tour along the binary tree,*

*(ii) if an agent is active in the latter then it is also active in the former.*

*Proof.* Recall that, by $\text{Inv}_t$ (iv) and Lemma 6, in each step of $\texttt{BS\_Jumps}(n, k)$ all active agents have the same token location. The next token location is some vertex in the subsequent part of the Eulerian tour. If an agent in the BS skips some vertices, it is because of feedback $0$ or $[n]$, and $\texttt{BS\_Jumps}(n, k)$ does it as well. Therefore, the token in the latter is never behind the token in the former. This proves point (i) of the lemma.

To prove point (ii), suppose, to the contrary, that for some configuration $K$ some agent $i \in K$ switches off earlier when executing BS than when executing $\texttt{BS\_Jumps}(n, k)$. By point (i), at that time its token location in the former is not after its token in the latter execution. By the query inclusion rule, the switch-off may happen only when the token is on the path to the root and the feedback is $[n]$. Let us call this vertex $v$. If the token in $\texttt{BS\_Jumps}(n, k)$ is at $v$ at that time, it also switches off and we obtain contradiction that finishes the proof. Otherwise, the token in $\texttt{BS\_Jumps}(n, k)$ is ahead of $v$ according to the Eulerian tour. If it was at $v$ earlier, then because it is the same configuration as in the other execution, agent $i$ would also get feedback $[n]$ to its query then and would switch off – leading again to contradiction ending the proof. Hence, the token must have passed across $v$ without choosing it for the token location, which means that $v$ was not ambiguous for configurations in $\mathcal{K}_\phi|_i$, where $\phi$ is the feedback sequence by that time. But since $K$ must also be in $\mathcal{K}_\phi|_i$, as $\phi$ is obtained during the execution for configuration $K$, it means that for every configuration $K' \in \mathcal{K}_\phi$ the feedback is the same as for $K$. However, the feedback for $K$ when query $Q_v$ is considered is $[n]$ – since agent $i$ switches off when configuration $K$ executes BS, and it happens only when $K \cap Q_v = \{i\}$. Hence, for every configuration $K' \in \mathcal{K}_\phi$, we have $i \in K'$ and no other leaf in $Q_v$ belongs to $K'$. Consequently, although the token in the $\texttt{BS\_Jumps}(n, k)$ execution could have passed through $v$ without choosing it, it needed to go directly to the first leaf on the Eulerian tour that belongs to every $\phi$-compatible configuration $K'$, which is $i$. Hence, $i$ would have also been selected in $\texttt{BS\_Jumps}(n, k)$ when the token was in its leaf, and thus switched off after that – this is a contradiction which concludes the last remaining case of the proof of Lemma 8. $\square$

It follows from Lemma 8(ii) that in an execution of $\texttt{BS\_Jumps}(n, k)$ without deviating agent, no agent starts its persistent inclusion – otherwise, it would become active for more than $4n$ step, which is longer than the length of the whole left-to-right Eulerian tour along the binary tree, which would violate Lemma 8(ii).

**Lemma 9.** $\texttt{BS\_Jumps}(n, k)$ *has maximum revealing time* $O(k \log(2n/k))$, *for any* $2 \le k \le n$.

*Proof.* The Binary Search algorithm BS has maximum latency of $\Theta(k + k \log(n/k)) = O(k \log(2n/k))$ Capetanakis [1979a]. It means that every agent has been revealed by that time when running BS. By Lemma 8(ii), no agent $i$ in any deviation-free execution of $\texttt{BS\_Jumps}(n, k)$ could finish later than in the execution of BS on the same configuration of agent (i.e., in any step $t$ it cannot be active in the former while being switched off in the latter) – thus, it switches off (which happens after successful selection, as grim-trigger sequence is not activated in deviation-free executions) in $O(k \log(2n/k))$ steps. Thus, the maximum revealing time is $O(k \log(2n/k))$. $\square$

For a given agent $i$, we call a vertex $v_i$ in the binary tree a *deviation point* for agent $i$ if $v_i$ is the first vertex $v$ in the left-to-right Eulerian tour such that

- (a) for all configurations $K$ containing agent $i$ and the corresponding executions of $\texttt{BS\_Jumps}(n, k)$ on them, the agent $i$ starts deviating not earlier than when its token location is at $v$ or after (according to the Eulerian tour), and

- (b) there is a configuration $K_i$ containing agent $i$ such that in the corresponding execution of $\texttt{BS\_Jumps}(n, k)$ the player $i$ starts deviating when its computed token location is at vertex $v$.

**Lemma 10.** *If $v_i$ is after the leaf $i$ (corresponding to agent $i$) according to the Eulerian tour, then there is no improving configuration.*

*Proof.* Indeed, according to the definition of $v_i$, in each configuration $K$ containing agent $i$, no deviation occurs until the token location is in $v_i$, but since token locations are consistent across agents before the deviation (see $\text{Inv}_t$ (iv) and Lemma 6, where $t$ is the deviation step number), agent $i$ succeeds before its token location reaches $v_i$, as $v_i$ is after leaf $i$ in the Eulerian tour. Otherwise, it would contradict $\text{Inv}_t$ (i-iii) and Lemma 6 with respect to configuration $K$, uniquely characterized before the token location $v_i$ (i.e., since $i$ is alone in some preceding subtree, there is a step $t'$ with

$\alpha_i[t']$ located at the root of some of such subtree, and feedback $\phi[t'] = 1$, which means that only agent $i$ in $K$ could have been revealed at that step). We conclude that the considered deviation does not improve revealing time of agent $i$ in any configuration. $\qquad\square$

It follows from Lemma 10 and the definition of $(n, k)$-AE that in such case we do not have to point out any worsening configuration. The next two lemmas cover the analysis of the remaining cases.

**Lemma 11.** *If $v_i$ is located on the path from the root to leaf $i$ (corresponding to agent $i$), then there is a worsening configuration.*

*Proof.* Assume that agent $i$ deviates (for the first time) when its token is in $v_i$ and $v_i$ is above $i$ or equal to $i$. Let $K$ be the configuration for which it happens. The query inclusion rule says that since $v_i$ is above leaf $i$ (corresponding to player $i$), agent $i$ should transmit in the algorithm; therefore, a deviation means that it chooses not to transmit at that step. If $K \cap Q_{v_i} = \{i\}$ then such a deviation automatically worsens the latency of agent $i$, as it would normally be revealed while because of deviation it is not. Note that there is at least one configuration $K \in \mathcal{K}_\phi|_i$ satisfying $K \cap Q_{v_i} = \{i\}$, as otherwise $Q_{v_i}$ would not be ambiguous (i.e., only feedbacks 0 or only feedbacks $\perp$ would have occurred) and thus $v_i$ would not have been chosen as token location by agent $i$.

Finally, observe that $v_i$ could be equal to $i$, but in this case a deviation (i.e., choosing not to be included) would obviously worsen the revealing time of agent $i$ whose turn is to be alone in the query according to the algorithm. $\qquad\square$

**Lemma 12.** *If $v_i$ is prior to the leaf $i$ (corresponding to agent $i$) according to the Eulerian tour, then there is a worsening configuration.*

*Proof.* If $v_i$ is located on the path from the root to leaf $i$, then the claim follows from Lemma 11. In the remainder, we assume that $v_i$ is not on that path – we show that there is a configuration that results in some agent with smaller id $j < i$ starting its grim-trigger sequence before successful selection of agent $i$, and therefore delaying it till step at least $4n + 1$ and thus worsening its revealing time (which would be $O(k \log(2n/k))$ if agent $i$ and all other agents were honest, by Lemma 9).

Consider the remaining case in which $v_i$ is located prior to the leaf $i$ on the Eulerian tour, but not on the path from the root to leaf $i$. By the definition of $v_i$, there is a configuration $K \in \mathcal{K}_\phi|_i$ during the execution of which, agent $i$ deviates when its token is in vertex $v_i$. By the inclusion rule, without deviation, agent $i$ would not be included in the query in such step; thus, after deviation, it is revealed. By the token update rule, $v_i$ is such that either it is an internal vertex and query $Q_{v_i}$ is ambiguous or it is a leaf $j$ belonging to all configurations $K' \in \mathcal{K}_\phi|_i$. In the latter case, there is a collision in this step (as the deviator $i$ is revealed), and since in the subsequent steps the token location computed by agent $j$ will be after leaf $j$ while $j$ has not been revealed, agent $j$ initiates persistent inclusion in the next $4n$ steps. During that time, no other agent could be revealed, and thus it worsens the revealing time of agent $i$. Note that in the above argument we used the fact that by the time of deviation, the algorithm is correct in the sense that all active agents have consistent token locations $v$ and are revealed if they are in query $Q_v$; this may not hold *after* the first deviation occurs in the execution.

In the remainder, we focus on the case when $v_i$ is an internal vertex of the binary tree. By $\mathrm{Inv}_t$ (v) and Lemma 6, where $t$ is the step when the deviation in token location $v_i$ happens, there is a configuration $K \in \mathcal{K}_\phi$ such that $|K \cap Q_{v_i}| = 1$. By $\mathrm{Inv}_t$ (i) and Lemma 6, for any $j \in Q_{v_i}$ there is a configuration $K_j \in \mathcal{K}_\phi$ such that $K_j \cap Q_{v_i} = \{j\}$. Note that $j \neq i$, as we consider now the case when leaf $i$ is not in the subtree of $v_i$. By $\mathrm{Inv}_t$ (iv) and Lemma 6, all active agents except the deviating agent $i$ use the same token location $v_i$ at the deviation time $t$ (we can apply this invariant because there were no deviations before, and the current token locations have been computed based on the preceding part of the execution without deviations). Therefore, in execution of any $K_j$, there will be a collision feedback in the considered step $t$, because the deviator $i$ will also be revealed. Thus, all active nodes choose the left-most child of $v_i$ as the next token location. If the deviating agent $i$ chooses again to deviate and is revealed, then we apply recursively the above argument and continue to the next step and move down the token location at all active and honest agents to the left-most child. If, however, agent $i$ chooses not to deviate, we restrict configurations to $K_j$ having $j$ in the subtree rooted at the (right-side) sibling of the current token – for such configurations, the feedback in this step will be 0 and the token at honest active agents automatically moves to the left-most child of the sibling. We continue the above recursive argument until we go down with the token location to a leaf, say $j$.

Now, if agent $i$ deviates then, it creates a collision with agent $j$, which makes agent $j$ to start persistent inclusion in the next step (as its token location passes its corresponding leaf $j$). It obviously worsens latency of agent $i$, as it could not be revealed for the next $4n$ steps.

The only remaining scenario is when agent $i$ does not transmit together with player $j$. Observe that the token has been moved from $v_i$ to $j$ through a sequence of recursive moves of token locations (described above), each next location at next level of the tree. Each of those intermediate token locations were ambiguous from perspective of honest agents (by straightforward inductive argument following from the recursive definition of next token location). At the same time, the recursive construction prevents agent $i$ from revealing itself for any $\phi$-compatible configuration. Since we do a recursive step at least once, and since each time the token moves down at least once, it also delays the maximum revealing time of agent $i$ by at least 1 comparing to the execution without deviation. Indeed, in the latter, the single agent at the first ambiguous vertex would transmit immediately and the token would jump immediately to the next ambiguous vertex in other parallel subtree. It completes the argument for the last case and the whole proof.

This finishes the proof of Lemma 12. □

Theorem 6.1 is a restatement of Theorem 3.1.

**Theorem 6.1.** *Algorithm* BS_Jumps$(n,k)$ *is an* $(n,k)$-*AE, for any* $2 \leq k \leq n$, *and has revealing time* $O(k \log(2n/k))$.

*Proof.* The maximum revealing time follows from Lemma 9. Being an $(n,k)$-AE follows from Lemmas 10 and 12. More precisely, for deviations that first occur after leaf $i$ on the Eulerian tour, Lemma 10 proves that there is no improving configuration; thus, according to the definition of $(n,k)$-AE, we do not need to prove existence of a worsening configuration. For other deviations, Lemma 12 proves that there is a worsening configuration. This completes the proof of the theorem. □

# 7 Other related work

Combinatorial Group Testing (CGT), to which we applied our game theoretical framework, is a classic area of learning theory, see the book Du et al. [2000]. It also has a connection with Contention Resolution problem and coding, see the following representative papers providing influential frameworks and solutions in both areas Capetanakis [1979a], Clementi et al. [2001], Kautz and Singleton [1964], Porat and Rothschild [2011]. The adversarial aspects of CGT was also considered in the classic (i.e., non game-theoretic) setting, c.f., Klonowski et al. [2022].

Shared channels model contention occurring in local-area networks, see Metcalfe and Boggs [1976], Chlebus [2001]. Bender et al. [2005] proposed to study broadcasting in shared channels with queue-free stations in the framework of adversarial queuing. Chlebus et al. [2012] introduced deterministic distributed broadcast performed by stations with queues in adversarial shared channels. Anantharamu et al. [2019] studied latency of broadcasting by deterministic algorithms in shared channels with adversarial packet injection. The meaning of "adversarial" in these papers refers to the way of how players are generated and is different to our adversary, who decides which $k \leq n$ players are present in a configuration of the game. These papers do not study game theoretic settings.

Contention resolution (CR) algorithms have been studied to help multiple users use efficiently a shared channel. Initially it was assumed that agents would respect the given algorithm. Assuming the channel with collision detection, Greenberg and Winograd [1985] prove a lower bound of $\Omega\left(\frac{k \log(n)}{\log(k)}\right)$ on the running time of any adaptive deterministic contention resolution algorithm. Capetanakis [1979a], Capetanakis [1979b] and Hayes [1978] independently found an adaptive, deterministic tree algorithm to solve the contention resolution problem with collision detection, which runs in $O(k + k \log(n/k))$ time. If we do not have collision detection, then Clementi et al. [2001, 2003] proved a lower bound of $\Omega(k \log(n/k))$ on the running time on any non-adaptive, therefore also any adaptive, deterministic algorithm for the CR problem.

Komlós and Greenberg [1985] proved that there exist non-adaptive CR algorithms (that do not use collision detection) with running time $O(k + k \log(n/k))$. This matched the lower bound in Clementi et al. [2001], but the proof of Komlós and Greenberg [1985] was existential using the probabilistic method. Later, Kowalski [2005] showed a polynomial time construction of CR protocol of length

$O(k \log^c n)$, for some constant $c > 1$, which employs partial selectors by Indyk [2002], Chlebus and Kowalski [2005], efficient dispersers (Ta-Shma et al. [2007]) and superimposed codes (Kautz and Singleton [1964] and Porat and Rothschild [2011]).

Acknowledgment-based, i.e., without collision detection, shared-channel algorithms, have been extensively studied in various communication problems, both deterministic and randomized Abramson [1970], Chlebus et al. [2012], De Marco and Stachowiak [2017], Hradovich et al. [2021], Komlós and Greenberg [1985], Kowalski [2005].

## 8   Discussion and further directions

We initiated a study of combinatorial group testing algorithms that are also adversarial equilibria – no agent/element has an incentive to deviate under presence of an adversary. Our solution in case of known size $k$ of the hidden set is almost optimal, in the sense that the price of stability is $O(\log k)$, i.e., the ratio between our result and the absolute lower bound on revealing time $\Omega(k \frac{\log n}{\log k})$ by Greenberg and Winograd [1985]. Closing this gap, and more importantly, devising an efficient equilibrium for less complex feedback function is an interesting direction. If negative results could be proved for deterministic solutions, then the question about efficient mixed equilibria (i.e., using randomized algorithms as strategies) would be interesting.

Another interesting question would be to try to close the gap between the bounds on maximum revealing time where the number $k$ of active agents is known, that is $O(k \log n)$, and the bounds where $k$ is unknown, which is $O(n^{k-1})$.

Other potential areas of application include: considering different feedbacks (e.g., full quantitative) and other related problems, e.g., discovering hidden multi-sets or transmissions of packets on a multiple-access channel without collisions. One may also consider weaker "risk awerseness" in the definition of AE, and how it influences efficiency of CGT.

Finally, it would be interesting to study connections of our games to repeated games, see, e.g., Osborne and Rubinstein [1994], where each agent has to transmit more than one packet.

## Acknowledgments and Disclosure of Funding

The work of Dariusz R. Kowalski was partially supported by the NSF grant 2131538. The work of Piotr Krysta has been partially supported by the Network Sciences and Technologies (NeST) initiative of the University of Liverpool.