# OpenReview forum: "Combinatorial Group Testing with Selfish Agents"
_NeurIPS.cc/2023/Conference — NeurIPS 2023 poster_

### Official Review · Reviewer_53TB · 2023-07-04

**Soundness:** 3 good
**Presentation:** 4 excellent
**Contribution:** 3 good
**Rating:** 7
**Confidence:** 3

**Summary:**

The paper studies Combinatorial Group Testing (CGT) problem in a game-theoretic setting. The setup consists of a set $[n]$ of $n$ agents, of which a set of $k$ agents, denoted by $K$ are `active'. The goal is to reveal their identities in the minimum time possible via set of queries. Each query $Q$ is a subset of $[n]$ and the feedback available to agents is information about $Q\cup K$. Authors present results using the notion of adversarial equilibrium (AE). Two scenarios are explored: one where the number of active agents (k) is known, and one where it is not. In the known scenario, they demonstrate AE strategies that achieve near-optimal revealing times $O(k \log(\frac{n}{k}))$. However, in the unknown scenario, the revealing time increases to the order of $n^{k-1}$ with a lower bound of $\Omega(n)$.

=====================================================================

EDIT:
I have improved the score from 6 to 7 after authors' response.

**Strengths:**

(+) The problem is very nicely and clearly introduced in Section 2. Also, the problem is quite interesting. The paper is well written, especially the technical aspects are well articulated and discussed.

(+) The algorithm for the case where $k$ is known (BS\_Jumps) and the analysis showing that the performance is ``close'' to the optimal is a sound contribution.

(+) CGT has been studied for decades. This new outlook leveraging game-theoretic ideas is indeed interesting (though this paper is not the first one to use it).

(+) Several interesting applications are identified and discussed.

**Weaknesses:**

(-) The paper essentially builds up on the ideas of Chionas et al. [2023]. It also seems like extension of their setups, which is not entirely a bad thing; however, it limits the extent of contributions. The main algorithmic contribution of the paper is basically the \emph{BS\_Jumps} algorithm. The other algorithms are either from the literature or are straight-forward.

(-) The feedback function used is not well motivated (at least this reviewer can not make a practical sense out of it). Please discuss  the implications of it.

(-) This paper offers theoretical results, which is good. However, there is no numerical evaluation/experiments section. I think a section dedicated to the empirical testing or validation of the proposed algorithms will surely enrich the paper. I think this is an important weakness of the paper.

(-) The case where $k$ is unknown is not thoroughly (at least not as thoroughly as the case with known $k$) discussed.
\end{itemize}

**Questions:**

(*) I do not understand the idea of `` selfishness'' of an agent. What constitutes the selfish behavior and what are the implications of that? Please clearly define and introduce it?

(*) In Section 2.1, is it true that none of the agents have information about the query generated at time $t$ (as one agent can deviate from the prescribed strategy). so, is it fair to say that ``algorithm'' is a central entity collecting the output of each agent's decision to be included in the query or not along with agents decision to deviate from the standard strategy or not. This central algorithm entity then creates a query.

(*) The authors present two new algorithms but don't provide enough information about their efficiency, besides mentioning the revealing time. For instance, are there any concerns about computational complexity, memory use, or scalability to larger problems?

**Limitations:**

(*) As I mentioned before, please discuss the motivation/limitations/implications of the specific feedback function (lines 117 -- 119).

(*) The authors mention various applications of CGT in the introduction, but it's unclear how well their AE model or algorithms would perform in these situations based on the information provided.

---

> ### Author Rebuttal · Authors · 2023-08-09
>
> **Weakness 1**: *The paper … builds up on ideas of Chionas et al. [2023]. It also seems like extension of their setups, which is not entirely bad thing; however, it limits the extent of contributions. The main algorithmic contribution of the paper is basically the BS_Jumps algorithm.*
>
>  Reply:
>
> We adapt their definition of Adversarial Equilibrium (AE) in our new setting with any algorithms  (strategies), including non-adaptive and adaptive ones, but Chionas et al. consider only non-adaptive strategies. This adaptation of AE requires new formalization of deviation -- deviation was fixed in advance (non-adaptive) in Chionas et al. while in our case, it could be decided by the deviating agent online. There are also minor differences in the problem setting, as they consider CR problem but we focus on general CGT. Adaptive strategies pose new significant challenges, discussed in the general rebuttal file, and as a result -- our algorithms, formulas and analysis are **entirely different**.
>
> **Weakness 2, Limitation 1**: *Feedback function used is not well motivated (at least this reviewer can not make a practical sense out of it). Please discuss the implications of it.*
>
> Reply:
>
> Our feedback function, defined in lines 117-119, is one of the simplest considered in  CGT, known in literature as ternary feedback or $(2, \log n)$-feedback in the generalized CGT nomenclature, cf. Klonowski et al. Generalized framework for group testing:... TCS 2022.
>
> If the query has a singleton intersection with the hidden set, then naturally feedback reveals the respective agent's ID. If the intersection is empty or has size at least $2$, then the feedback is empty or clash, respectively. We chose this feedback for presentation of our ideas because of its simple nature (eg, no need to "decode'' the ID of singletons).
>
> Probably even more popular feedback is when instead of ID of the singleton, a value $1$ or range $\ge 1$ (so called "beeping") is revealed.
>
> We show (Suppl. Materials Sec. 5) how to extend our results to such weaker CGT feedbacks and to the setting used in the CR problem.
>
> Thus, our results cover a wider range of natural, well motivated CGT and CR settings.
>
> **Weakness 3**: *The case where $k$ is unknown is not thoroughly (at least not as thoroughly as the case with known) discussed.*
>
> Reply:
>
> The case of unknown $k$ is less discussed, as we prove a strong lower bound in this case - we prove that any CGT+AE algorithm in this case has revealing time at least $\Omega(n)$.
>
> We also design and analyze an algorithm which is polynomial in $n$ for any constant $k$; see also our response to Q1 of Reviewer X2Ux regarding technical challenges.
>
> It is interesting, though, to explore cases between knowing and not knowing at all the number of active agents, eg, having partial knowledge about $k$, an upper bound on $k$, or even consider the number of active agents as a random variable and devise bounds for the distribution of this number.
>
> **Q1:** *I do not understand the idea of "selfishness" of an agent. What constitutes the selfish behavior and what are the implications of that? Please clearly define and introduce it?*
>
> Reply:
>
> In our setting, each agent's goal is to minimize its revealing time. Selfish behavior means that an agent could deviate from what a query says, see lines 169-176, in order to reduce its revealing time. As we prove that our algorithms are AE, no agent will deviate unilaterally without a threat of worsening their revealing time. An implication of selfish behavior in equilibrium is that the respective metric of global efficiency is compromised compared to the cooperative setting. This is quantified by Price of Anarchy, and the loss in efficiency is depicted in Table 1.
>
> **Q2:** *In Sec. 2.1, is it true that no agent has information about query generated at time $t$ (as one agent can deviate from the prescribed strategy). so, is it fair to say that "algorithm" is a central entity collecting output of each agent's decision to be included in the query or not along with agents decision to deviate from standard strategy or not. This central algorithm entity then creates a query.*
>
> Reply:
>
> Reviewer’s intuition is correct, our only comment is that we use word "algorithm" in association with strategies (or computing them) at each agent, while the "centralized entity" mentioned by Reviewer is actually the **feedback function**. It was defined within CGT setting (from line 115), but we will add a reminder after line 168 that: after each agent decides if it is present in the query or not, the feedback function’s outcome is computed based on the agents' decisions and communicated to all agents.
>
> **Q3:** *The authors present two new algorithms but don't provide enough information about their efficiency, besides mentioning the revealing time. For instance, are there any concerns about computational complexity, memory use, or scalability to larger problems?*
>
> Reply:
>
> The main efficiency metric is the revealing time (query complexity), i.e., number of game’s rounds. Total communication complexity can be derived as a function of protocol’s revealing time. In each round, each active agent communicates a single bit. The ternary feedback can be encoded by $\log n$ bits. Let $T$ be the protocol’s revealing time. Thus, total communication complexity of a protocol is $(k+\log n) \cdot T$ bits. The total number of local steps during the game can be upper bounded by $\log n \cdot T$, as each round of algorithm BS_Jumps requires updating token, which is a logarithmic operation (see Suppl. Materials, Sec. 3). The memory needed in each agent is logarithmic, due to simple recursion in main algorithm and token update and a constant number of variables. This is assuming that basic arithmetic operation on values, with logarithmic bit representation, is done in single local step, and these numbers are stored in a memory unit.
>
> **Limitation 2** applications addressed in the general rebuttal file.

---

> > ### Comment · Reviewer_53TB · 2023-08-17
> > **Thanks for the clarification**
> >
> > Thank you for the clarification. I believe authors addressed my question in sufficient details, including more information about feedback function, explanation of terms, distinction and positioning of works in the contexts of known works.
> >
> > The only remaining "concern" is the weakness I mentioned in the earlier review. "This paper offers theoretical results, which is good. However, there is no numerical evaluation/experiments section. I think a section dedicated to the empirical testing or validation of the proposed algorithms will surely enrich the paper. I think this is an important weakness of the paper."
> >
> > Authors have not commented on it. I will be happy to increase my score after authors comment on that. Please note that I just would like to know why no experiments are included, or why there is no good need of them here? Thanks.

---

> > > ### Author Response · Authors · 2023-08-18
> > > **Numerical evaluation/experiments**
> > >
> > > We thank to the reviewer for the comments. Here are our explanations about the experiments: We have an ongoing cooperation with a blockchain company which runs the global blockchain Tezos, where we proposed to them our algorithm BS_Jumps as a fairness mechanism for their PoS (Proof of Stake)-based consensus protocol (as described in Section 5.4 of our Supplementary Materials). They are currently considering the implementation and experimenting with our algorithm on their platform but their policy and timing did not allow us yet to include these in the NeurIPS submission.

---

> > > > ### Comment · Reviewer_53TB · 2023-08-18
> > > > **Thanks**
> > > >
> > > > Thanks for the explanation. It is great that you are pursuing the applications of the work with an industry partner. However, nothing stops from doing an experimental/numerical evaluation. You do not have to showcase the experiments done with the industry partner. However, a numerical evaluation on other/synthetic data could be easily done without constraints. Nevertheless, considering the authors' satisfactory responses to more important concerns, I have improved the score. Good luck.

---

### Official Review · Reviewer_LHGX · 2023-07-06

**Soundness:** 3 good
**Presentation:** 3 good
**Contribution:** 2 fair
**Rating:** 7
**Confidence:** 2

**Summary:**

This submission applies and extends a very recent game theoretic perspective, introducing in particular a notion of strategy equilibrium called adversarial equilibrium, on contention resolution games to the more general setting of combinatorial group testing. In combinatorial group testing, a (typically small) group must be revealed among a large set of candidates by feedback to certain queries. In this game theoretic setting the small group is considered as the set of players who execute strategies to include or remove themselves from the queries prescribed by an algorithm given on input. Adversarial equilibrium captures the notion that, no deviation from a strategy allows a to not increase and to strictly decrease their revealing time for all adversarial and one adversarial choice of the small set. It also provides strategies for agents which satisfy an adversarial equilibrium and achieve almost the best possible bound (the lower bound is known and independent from the considered game theoretic model) on the latest revealing time of a player in the small group when the size of the small group is known, and less tight upper and lower bounds for the latest revealing time of a player achievable by adversarial equilibrium strategies.

**Strengths:**

The combinatorial group testing problem is highly relevant, also in the context of machine learning and hence fits the scope of NeurIPS well.
The contribution claims to generalise some recent results accepted for publication at a strong venue (unfortunately the proceedings seem not to be available yet and hence I was not able to have a closer look at it).
The quality of the technical writeup is good and I did not spot any flaws.

**Weaknesses:**

To me the motivation of requiring adversarial equilibria or even considering CGT as games played by the small set that should be revealed is not sufficiently supported in my opinion. In particular I would appreciate a discussion of settings of CGT in which this is natural to consider (the applications provided in the supplementary material are a step in this direction but I would ask even more explicitly why one would be interested in designing AE strategies for blockchain mechanisms; for CR this seems a bit more clear to me but this is also somewhat unsurprising as this is the context in which AE was introduced) as well as a more explicit comparison of how `adversarial aspects' have been addressed in CGT previously.

-L76: it is not completely clear to me, what `usefulness' is demonstrated here. By this I do not mean to say that the presented results are uninteresting but rather that I do not understand how they make a point of any benefit behind requiring AE as it is even presented for unknown k that this comes at the necessary cost of worse latest revealing times.

**Questions:**

Can you clarify the above points relating to the motivation behind taking a game theoretic perspective and requiring adversarial equilibrium for CGT?

**Limitations:**

No concerns.

---

> ### Author Rebuttal · Authors · 2023-08-09
>
> We thank the reviewer for valuable feedback. Below we give our answers. If the reviewer believes that we have addressed adequately all of her/his concerns, we would very much appreciate if she/he reconsiders upgrading the score for our paper.
>
> **Question 1:**
> *To me the motivation of requiring adversarial equilibria or even considering CGT as games played by the small set that should be revealed is not sufficiently supported in my opinion. In particular I would appreciate a discussion of settings of CGT in which this is natural to consider (the applications provided in the supplementary material are a step in this direction but I would ask even more explicitly why one would be interested in designing AE strategies for blockchain mechanisms; for CR this seems a bit more clear to me but this is also somewhat unsurprising as this is the context in which AE was introduced) as well as a more explicit comparison of how "adversarial aspects" have been addressed in CGT previously.*
>
> Response:
>
> Our CGT+AE framework could be applied to **assure fairness** in any application of the CGT which assumes some autonomy of the elements. For instance, one could consider new blockchain mechanisms in which miners compete to add their blocks to the blockchain in a fair way. To do it, they could pick or be assigned with a random ID, and our new framework assures that all blocks will be efficiently added, in a random order (as no miner has any incentive to deviate the search going through random IDs).
>
> Another application that uses the full CGT setting is where we are given a distributed database with $n$ selfish servers, where some $k$ of those servers hold $k$ pieces of information that a user is looking for in the database. In the beginning, the IDs of these servers are not known to the user (in this sense, we could model it as adversarial choice). Each of the $k$ servers wants to have its information to be found ("released") as early as possible (because it can use its resources to serve the next query when it is free and does not need to wait to "release" its information, or could be even awarded for promoting the information stored on it). A possible approach could give random ids to the servers before running a CGT+AE search, thus also ensuring fairness in the treatment of the servers, unless they would deviate (but that is prevented, or at least discouraged, by the solution being an AE).
> ​
> Similar technique could be applied to the master-workers systems, including Distributed/Decentralized/Federated ML and AI. In order to **avoid biases** when processing the results of the jobs in some order, the master could require the workers who are ready (in our model -- active) to submit their results in random order. To do it, the master attaches random IDs to the jobs sent to workers, and asks those of them who are ready to execute the CGT protocol which is AE. The CGT part assures that all work is collected. Random IDs of jobs assure random order, provided no worker deviates. AE part assures that no matter which set of workers is ready, none of them is incentivized to deviate.
> ​
> There are also several other applications of CGT considered in recent ML/AI publications, to different types of searches, string mining, etc. cf.,
> ​
> - J. Engels, B. Coleman, and A. Shrivastava. Practical near neighbor search via group testing. NeurIPS 2021
> ​
> - D.R. Kowalski and D. Pajak. Light agents searching for hot information. IJCAI 2022
> ​
>
> We will include these applications in the proceedings version of our paper (as typically 1 extra page is available in the proceedings) if the paper is accepted.
>
>
> **Question 2:**
> *-L76: it is not completely clear to me, what "usefulness" is demonstrated here. By this I do not mean to say that the presented results are uninteresting but rather that I do not understand how they make a point of any benefit behind requiring AE as it is even presented for unknown k that this comes at the necessary cost of worse latest revealing times.*
>
> Response:
>
> Apologies for a confusion. We used the word "usefullness" in the wrong context. What we meant here is that our added notion of AE does make a difference comparing to the classical CGT (and related settings, such as CR) because it leads to different algorithms, techniques and performance bounds.

---

> > ### Comment · Reviewer_LHGX · 2023-08-18
> >
> > Thank you for your response. All points raised by weaknesses are addressed to my satisfaction and I will raise my score to a recommend acceptance.

---

### Official Review · Reviewer_X2Ux · 2023-07-07

**Soundness:** 3 good
**Presentation:** 4 excellent
**Contribution:** 3 good
**Rating:** 6
**Confidence:** 3

**Summary:**

This paper introduces a game theoretic framework in the context of combinatorial group testing. In detail, in a population of $n$  elements, there are $k<n$ agents that are considered active (e.g. they are positive to some disease) and the goal is to use queries on subsets of the $n$ elements to reveal who those $k$ agents are. These active agents can strategize, i.e., choose whether they are included in the queries used to reveal the subset of active elements.

The authors introduce a modified notion of adversarial equilibrium that allows adaptive queries (i.e. queries that can depend on previous queries).

**Strengths:**

* Clear and interesting outline of previous work.
* Accessible language, self-contained definitions, and overall clear presentation of background and novel ideas.
* The game theoretic setting seems interesting, natural, and well-motivated.

**Weaknesses:**

The main weakness is the fact that the main results have to do with an a priori known number of active elements $k$.

**Questions:**

* What would be the main challenge in getting similar results for an unknown number $k$?
* Why do you not consider non-active elements as non-strategizing?
* Could you clarify the connection between CR and CGT?

**Limitations:**

The limitations are, to the best of my understanding, adequately discussed.

---

> ### Author Rebuttal · Authors · 2023-08-09
>
> We thank the reviewer for valuable feedback. Below we give our answers. If the reviewer believes that we have addressed adequately all of her/his concerns, we would very much appreciate if she/he reconsiders upgrading the score for our paper.
>
> **Weakness:**
> *The main weakness is the fact that the main results have to do with an a priori known number of active elements.*
>
> Response:
>
> We prove that, it is the nature of the problem that the lack of any knowledge of $k$ yields high complexity.
> More precisely, we prove that if $k$ is unknown, no algorithm can guarantee a revealing time smaller than $\Theta(n)$ -- please see our lower bound; in this case we show an algorithm with the revealing time of $O(n^k)$. Hence, for unknown $k$ the problem is provably hard. On the other hand, this question of the reviewer opens an interesting avenue for further work. That is, we can ask what happens if we have some partial knowledge about $k$, for instance, a rough upper bound on $k$, e.g., $k \leq k^*$ for some $k^*$ known to the algorithm? In this case, could one prove an upper bound on the revealing time that is, for instance, polynomial in $k^*$ and $\log(n)$ ?
>
> Another opportunity for efficient algorithms would be a model with extended feedback function, e.g., returning some approximate or even exact size of the intersection of the query and the hidden set (so called Quantitative Group Testing). However, one would need to study if there are any beneficial deviations due to such extended feedback received in each round of the game.
>
>
> **Question 1:**
> *What would be the main challenge in getting similar results for an unknown number $k$?*
>
>
> Response:
>
> If $k$ is unknown, the adversary has even more power, since it not only selects the configuration but also the size, and the agents are agnostic to it.
> Our current approach is, roughly speaking, to query complements of all possible configurations for increasing values of $k-1$ (starting from $k-1=1$), so that when the considered size of a configuration is correct, any deviation leads to discoverable effect and a "punishment strategy" could be applied by non-deviating agents. Finding a way to avoid addressing all configurations of a given size in separate queries, and thus (substantially) reducing the number of queries, seems to be the main challenge in the setting with unknown $k$. We formally show that this setting is more difficult than the one with known $k$
> by proving a lower bound of $\Omega(n)$ on the revealing time.  We remind that in BS_Jumps$(n,k)$ we achieve equilibrium with low latency because of the knowledge of the $(k-1)$'st revelation -- this is not possible in the case of unknown $k$.
>
> **Question 2:**
> *Why do you not consider non-active elements as non-strategizing?*
>
> Response:
>
> The nature of CGT splits agents into $k$ active agents and the remaining $n-k$ non-active agents. However, at the beginning of the game no agent knows which other agents are active and therefore could strategize. The set of active agents is being revealed as the game proceeds. But we cannot assume that from the perspective of any (active) agent, some other agent, say $i$, is non-strategizing, because that agent may not know whether $i$ is active until the whole set of active agents is revealed.
> That is, the non-active agents cannot be assumed to be non-strategizing, because the protocol (AE) must be prepared for ANY of the $n$ agents to be chosen (by the adversary) as active, and therefore -- prevent its potential deviations.
>
> **Question 3:**
> *Could you clarify the connection between CR and CGT?*
>
> Response:
>
> Formally speaking, CR is a communication problem in which each player has a packet to send successfully (i.e., alone) on the multiple-access channel to which it is connected (together with other agents). CGT is a learning problem in which a hidden set must be discovered by asking queries and receiving feedback.
> The main conceptual similarity between CR and CGT is the environment -- agents decide to transmit (equiv., be included in the query) or not and they receive feedback from the channel (equiv., broker of the CGT process -- the feedback function). Differences are in details, as CR problem -- as a communication problem -- may adopt communication features, such as acknowledgments (i.e., feedback received only by a single transmitter), possibility of sending actual information to be received by others in case of successful/alone transmission (apart of the feedback), possibility of performing fake/jamming transmissions (even if it is an alone transmission, it does not fulfill the goal of CR because the actual packet has not been transmitted), etc.
> Another subtle difference is that in CR each player has to transmit alone at least once in order to fulfill its goal, which in CGT we may sometimes reveal the hidden set based on feedback analysis, even if not every active element occurred in a singleton intersection with a query.
> We also provide some intuitions about transformation from CGT to CR in the Supplementary Materials Section 5; we could copy it to the main part, space permitting.

---

> > ### Comment · Reviewer_X2Ux · 2023-08-16
> > **Thanks**
> >
> > Thank you for the detailed responses. I would like to raise my score to 6.

---

### Official Review · Reviewer_YF8o · 2023-07-10

**Soundness:** 3 good
**Presentation:** 3 good
**Contribution:** 2 fair
**Rating:** 6
**Confidence:** 1

**Summary:**

In this paper, the authors study the problem of combinatorial group testing in a new framework, when the elements to be found are selfish players. The authors propose an algorithm such that no agent is willing to deviate from the algorithm strategy (being in this way in an adversarial equilibrium).

**Strengths:**

- The paper is well-written and very clear. The graphical example the authors provide helps in understanding the mechanism beyond the algorithm.

- The authors provide lower bounds on the novel problem setting and an algorithm to solve the problem.

- I checked the proofs and the results seem sound and well-explained.

**Weaknesses:**

- It is not very clear to me which is the application of this approach. What kind of application cannot be modeled with classical CGT and they can be modeled with this multi-agent setting? In which cases do these selfish agents appear? I saw that there are some examples in the appendix, but it would be helpful to spend more time describing them.

- I suggest moving to the main paper an intuition for the presented lower bounds. It is always helpful to understand the hardness of the problem.

- What are the main technical challenges with respect to Chionas et al.?

**Questions:**

See weaknesses.

**Limitations:**

No potential negative societal limitations.

---

> ### Author Rebuttal · Authors · 2023-08-09
>
> We thank the reviewer for valuable feedback. Below we give our answers. If the reviewer believes that we have addressed adequately all of her/his concerns, we would very much appreciate if she/he reconsiders upgrading the score for our paper.
>
> **Question 1:**
> *It is not very clear to me which is the application of this approach. What kind of application cannot be modeled with classical CGT and they can be modeled with this multi-agent setting? In which cases do these selfish agents appear? I saw that there are some examples in the appendix, but it would be helpful to spend more time describing them.*
>
> Response:
>
> Our CGT+AE framework is not actually competing, in terms of applications, with the classical CGT, but it is rather an add on. It could be applied to **assure fairness** in any application of the CGT which assumes some autonomy of the elements. For instance, one could consider new blockchain mechanisms in which miners compete to add their blocks to the blockchain in a fair way. To do it, they could pick or be assigned with a random ID, and our new framework assures that all blocks will be efficiently added, in a random order (as no miner has any incentive to deviate the search going through random IDs).
>
> Another application that uses the full CGT setting is where we are given a distributed database with $n$ selfish servers, where some $k$ of those servers hold $k$ pieces of information that a user is looking for in the database. In the beginning, the IDs of these servers are not known to the user (in this sense, we could model it as adversarial choice). Each of the $k$ servers wants to have its information to be found ("released") as early as possible (because it can use its resources to serve the next query when it is free and does not need to wait to "release" its information, or could be even awarded for promoting the information stored on it). A possible approach could give random ids to the servers before running a CGT+AE search, thus also ensuring fairness in the treatment of the servers, unless they would deviate (but that is prevented, or at least discouraged, by the solution being an AE).
> ​
> Similar technique could be applied to the master-workers systems, including Distributed/Decentralized/Federated ML and AI. In order to **avoid biases** when processing the results of the jobs in some order, the master could require the workers who are ready (in our model -- active) to submit their results in random order. To do it, the master attaches random IDs to the jobs sent to workers, and asks those of them who are ready to execute the CGT protocol which is AE. The CGT part assures that all work is collected. Random IDs of jobs assure random order, provided no worker deviates. AE part assures that no matter which set of workers is ready, none of them is incentivized to deviate.
> ​
> There are also several other applications of CGT considered in recent ML/AI publications, to different types of searches, string mining, etc. cf.,
> ​
> - J. Engels, B. Coleman, and A. Shrivastava. Practical near neighbor search via group testing. NeurIPS 2021
> ​
> - D.R. Kowalski and D. Pajak. Light agents searching for hot information. IJCAI 2022
> ​
>
> We will include these applications in the proceedings version of our paper (as typically 1 extra page is available in the proceedings) if the paper is accepted.
>
>
> **Question 2:**
> *I suggest moving to the main paper an intuition for the presented lower bounds. It is always helpful to understand the hardness of the problem.*
>
> Response:
>
> If our paper gets accepted, we will add these aspects to the main body of the paper, as well as the details of the above application to blockchain. (NeurIPS  typically allows one more page in the proceedings.)
>
> **Question 3:**
> *What are the main technical challenges with respect to Chionas et al.?*
>
> Response:
>
> The main challenge arises from the fact that our work considers adaptive strategies while the one by Chionas et al. considers only non-adaptive strategies.

---

> > ### Comment · Reviewer_YF8o · 2023-08-21
> >
> > I would like to thank the authors for the detailed rebuttal. I followed the discussion with the other reviewers and carefully read your answer. Due to this, I continue to recommend the acceptance of the paper.

---

### Author Rebuttal · Authors · 2023-08-09

We would like to thank the reviewers for their effort and valuable feedback. Below we address the main two requests for providing more applications and describing technical challenges comparing to Chionas et al. IJCAI 2023. Other questions we address as individual rebuttals to the reviewers.

**More details on applications of the CGT+AE framework.**


Response:
Our CGT+AE framework is not actually competing, in terms of applications, with the classical CGT, but it is rather an add on. It could be applied to **assure fairness** in any application of the CGT which assumes some autonomy of the elements. For instance, one could consider new blockchain mechanisms in which miners compete to add their blocks to the blockchain in a fair way. To do it, they could pick or be assigned with a random ID, and our new framework assures that all blocks will be efficiently added, in a random order (as no miner has any incentive to deviate the search going through random IDs).

Another application that uses the full CGT setting is where we are given a distributed database with $n$ selfish servers, where some $k$ of those servers hold $k$ pieces of information that a user is looking for in the database. In the beginning, the IDs of these servers are not known to the user (in this sense, we could model it as adversarial choice). Each of the $k$ servers wants to have its information to be found ("released") as early as possible (because it can use its resources to serve the next query when it is free and does not need to wait to "release" its information, or could be even awarded for promoting the information stored on it). A possible approach could give random ids to the servers before running a CGT+AE search, thus also ensuring fairness in the treatment of the servers, unless they would deviate (but that is prevented, or at least discouraged, by the solution being an AE).

Similar technique could be applied to the master-workers systems, including Distributed/Decentralized/Federated ML and AI. In order to **avoid biases** when processing the results of the jobs in some order, the master could require the workers who are ready (in our model -- active) to submit their results in random order. To do it, the master attaches random IDs to the jobs sent to workers, and asks those of them who are ready to execute the CGT protocol which is AE. The CGT part assures that all work is collected. Random IDs of jobs assure random order, provided no worker deviates. AE part assures that no matter which set of workers is ready, none of them is incentivized to deviate.

There are also several other applications of CGT considered in recent ML/AI publications, to different types of searches, string mining, etc. cf.,

- J. Engels, B. Coleman, and A. Shrivastava. Practical near neighbor search via group testing. NeurIPS 2021

- D.R. Kowalski and D. Pajak. Light agents searching for hot information. IJCAI 2022

We will include these applications in the proceedings version of our paper (as typically 1 extra page is available in the proceedings) if the paper is accepted.


**Main (technical) challenges with respect to the paper by Chionas et al.**

Response:
The main challenge arises from the fact that our work considers all possible strategies (adaptive and non-adaptive) while the one by Chionas et al. considers only non-adaptive strategies:
- Thus, on one hand, adaptiveness  considered in our work allows building a wider class of strategies that adapt to the feedback history of the game, possibly being much better and faster revealing the hidden set. This opens additional opportunities of designing more efficient algorithms, but makes the proofs of lower bounds more difficult (as they need to hold for any feasible algorithm).

- However, on the other hand -- it is more difficult to assure that such adaptive strategies form an equilibrium, comparing to the analysis of non-adaptive strategies in Chionas et al. This is because there are much more deviating adaptive strategies than non-adaptive ones -- i.e., the deviating agent can decide about deviation online, even after revealing part of the hidden set, while non-adaptive deviating strategy must fix the changes in its strategy in advance (and they stay the same during the whole game, regardless of what are the other elements in the actual hidden set). In particular, in our analysis, we could not just fix a deviating strategy and consider any hidden set (as in Chionas et al.), but we also need to take into account deviations occurring after revealing some elements from the hidden set during the game.

Other challenges, though not as critical as above, arise from subtle differences between the Contention Resolution problem (focused on in the other work) and CGT (focused on in our work), for instance, we do not assume the feature analogous to "jamming transmissions" which was helpful in case of non-adaptive CR to enforce "punishment mechanism".
See also our response to Reviewer X2Ux for more details on differences between CGT (considered in this work) and CR problem (studied by Chionas et al.).

As a result of the abovementioned new challenges, our algorithms and lower bounds are **entirely different** (in terms of design, formulas and analysis) from the ones in Chionas et al.

---

### Decision · Program_Chairs · 2023-09-21

**Decision:**

Accept (poster)

**Comment:**

In this paper the author introduce a novel problem of combinatorial group testing as follows: There are n agents out of which k (k < n) are belong to the group of target (e.g. they all have a target featuer), and the goal is to use efficient number of queries on subsets of the n agents to correctly identify the k target agents. The novel setting is that the k target agents can strategize, i.e., choose whether they are included in the queries.

The main contribution of this paper is a modified notion of adversarial equilibrium that allows adaptive queries (i.e. queries that can depend on previous queries) and the analysis of this new equilibrium.

After a productive rebuttal phase, the only remaining concern from the reviewers is the lack of experiments. However, I don't this this is a major issue as the paper is theoretical. Hence I recommend accepting this paper.